# Placental secretome characterization identifies candidates for pregnancy complications

Tina Napso[1], Xiaohui Zhao [1], Marta Ibañez Lligoña [1], Ionel Sandovici [1,2], Richard G. Kay[3], Amy L. George [3], Fiona M. Gribble [3], Frank Reimann [3], Claire L. Meek[3], Russell S. Hamilton [1,4] & Amanda N. Sferruzzi-Perri [1✉]

Alterations in maternal physiological adaptation during pregnancy lead to complications, including abnormal birthweight and gestational diabetes. Maternal adaptations are driven by placental hormones, although the full identity of these is lacking. This study unbiasedly characterized the secretory output of mouse placental endocrine cells and examined whether these data could identify placental hormones important for determining pregnancy outcome in humans. Secretome and cell peptidome analyses were performed on cultured primary trophoblast and fluorescence-activated sorted endocrine trophoblasts from mice and a placental secretome map was generated. Proteins secreted from the placenta were detectable in the circulation of mice and showed a higher relative abundance in pregnancy. Bioinformatic analyses showed that placental secretome proteins are involved in metabolic, immune and growth modulation, are largely expressed by human placenta and several are dysregulated in pregnancy complications. Moreover, proof-of-concept studies found that secreted placental proteins (sFLT1/MIF and ANGPT2/MIF ratios) were increased in women prior to diagnosis of gestational diabetes. Thus, placental secretome analysis could lead to the identification of new placental biomarkers of pregnancy complications.

[1] Centre for Trophoblast Research, Department of Physiology, Development and Neuroscience, University of Cambridge, Cambridge, UK. [2] Metabolic Research Laboratories, MRC Metabolic Diseases Unit, Department of Obstetrics and Gynaecology, The Rosie Hospital, Cambridge, UK. [3] Wellcome-MRC Institute of Metabolic Science, Addenbrooke's Hospital, Cambridge, UK. [4] Department of Genetics, University of Cambridge, Cambridge, UK. ✉email: ans48@cam.ac.uk

The placenta forms the functional interface between the mother and fetus that is essential for fetal development and growth during pregnancy. It is responsible for secreting a plethora of endocrine mediators that induce local and systemic changes in the mother to enable fetal nutrient and oxygen transfer and prevent immunological rejection of the fetus[1]. Aberrant placental function can lead to insufficient or inappropriate adaptations in maternal physiology, with consequences for pregnancy outcome and with immediate and lifelong impacts on the health of both the mother and child. Indeed, placental malfunction is a leading cause for the development of pregnancy complications, such as preeclampsia (PE), gestational diabetes mellitus (GDM), and intrauterine growth restriction (IUGR). Combined, these complications affect up to 6–8% of pregnancies in the UK (https://www.gov.uk/government/statistics/birth-characteristics-england-and-wales-2014). Typically, these complications are diagnosed in the second or even the third trimester of gestation, after the complication has already manifested. Moreover, current diagnosis methods, namely blood pressure and proteinuria evaluation for PE, oral glucose tolerance test (OGTT) for GDM, and uterine fundal height and ultrasound measures for IUGR are performed at a specific time/s in gestation and the development of the complication may not be detected in some cases. Therefore, the identification of novel placental biomarkers for earlier detection and improved diagnosis of pregnancy complications is highly desirable. Moreover, the illumination of placental biomarkers may aid in the design of novel therapeutic targets for pregnancy complications.

The notion that placental biomarkers may provide diagnostic or prognostic value for pregnancy complications is a long-standing and supported concept. For instance, detection of the placental hormone, chorionic gonadotropin is used to confirm pregnancy, reduced levels of pregnancy-associated plasma protein-A in the maternal circulation are predictive of IUGR and PE[2] and an imbalance in placental derived angiogenic regulators, like soluble fms-like tyrosine kinase 1 (sFLT1) and placental growth factor (PlGF) can be predictive of PE[3]. However, studies in experimental animals have demonstrated that the production of many other protein hormones by the placenta could also be important in determining pregnancy outcome[1,4]. In rodents, placental lactogens/prolactins (PL/PRL), growth hormone (GH), and insulin-like growth factor 2 (IGF2) modulate maternal insulin and glucose levels during pregnancy and perturbed expression of these proteins by the placenta have been associated with GDM and abnormal fetal growth in humans. The placenta also produces a wide variety of cytokines throughout pregnancy, which contribute to the low-grade systemic inflammation and induction of maternal insulin resistance that normally occurs in the second half of pregnancy and some data suggest that placental cytokine production is aberrant in women with poor pregnancy outcomes like PE, GDM, and IUGR[5]. Additionally, the placenta secretes inhibins, activins, and relaxins, which aid in the adaptation of the endocrine, renal, circulatory, and immune systems of the mother during gestation[1]. Finally, the placenta secretes proteases, inhibitors of peptidases, binding proteins and soluble forms of receptors for steroids, growth factors, cytokines, and circulating factors, like lipoproteins, which contribute to the pleiotropic endocrine regulation of maternal physiology during gestation and show some predictive value for conditions like IUGR[6–8]. Thus, there is likely a constellation of protein mediators secreted by the placenta that facilitate maternal adaptations and ensure adequate fetal growth, required for a healthy pregnancy outcome.

Transcriptomic analyses have informed on the repertoire of hormones expressed by the human placenta in healthy and complicated pregnancies[9–12]. However, these studies are conducted mainly on samples obtained at delivery and involve analysis of pieces of placenta tissue, which is heterogeneous in nature and includes trophoblast, vascular, stromal, and specialized immune cell types. Moreover, as powerful a tool it may be, transcriptomic analyses on their own may not be sufficient to identify the protein hormones secreted by the placenta. This is because genes can be subjected to posttranscriptional and posttranslational regulatory mechanisms, such as alternative splicing, folding, transport, localization, degradation, and secretion. Thus, analysis of the secretome, the complete list of proteins secreted by the placenta, would be invaluable for identifying placental biomarkers of maternal and fetal wellbeing that could be altered prior to the manifestation of a pregnancy complication.

The mouse is a valuable species for defining the placental secretome. This is because hormone secretion by the placenta is principally performed by trophoblast endocrine cells that are conveniently arranged into a structure, termed the junctional zone. The junctional zone is also discrete from, and forms under distinct genetic instruction to, the labyrinthine zone, which performs substrate transport function in the mouse placenta. This is in contrast to humans, where the endocrine and transport functions of the placenta are carried out by the same region/cell type, the syncytiotrophoblast (STB), preventing the specific, sole examination of placental hormone production. The mouse also offers the key advantage that tools to selectively modify and isolate endocrine cells in the placenta are now available[13]. Moreover, despite some variations between mice and humans, many mouse-specific hormone genes are structurally similar to those in the human and perform similar functions (e.g., PRL/PL and GH genes)[1]. Furthermore, many gene and protein networks regulating placental development and function overlap in the two species[14,15].

Herein, we first established a method for obtaining primary cultures of mouse placental endocrine cells from which the cells and secretory output could be collected. As a complementary approach, we also employed fluorescence-activated cell sorting (FACS) to isolate endocrine cells from the placenta of mice. We used mass spectrometry to unbiasedly identify the proteins in our different mouse placental endocrine cell preparations and applied a bioinformatic pipeline to refine a placental secretome map. We then overlaid our placental secretome map to a compilation of RNA/protein expression databases publicly available for the human placenta in women with pregnancy complications, including PE, GDM, and IUGR to identify secreted placental proteins that could be clinically important. As a proof of concept, we quantified the abundance of four secreted placental protein candidates (sFLT1, MIF, ANGPT2, and IGF2) and their ratios to one another in blood samples taken from women who had uneventful/healthy pregnancy outcomes and those who developed GDM (both populations were normotensive). We found that sFLT1 was altered in abundance in women with GDM and moreover, the ratios of sFLT1 or ANGPT2 to MIF were altered in the first trimester of pregnancy in women who went on to develop and be diagnosed with GDM in the second trimester of pregnancy. Finally, we identified several transcription factors (TFs) that are predicted to be important for controlling endocrine function of the placenta and determining pregnancy outcome. Our methodology and placental secretome map may be useful in identifying additional placental biomarkers for pregnancy complications.

## Results

We first wanted to establish a method for obtaining primary cultures of mouse placental endocrine cells from which secretory output could be collected and unbiasedly characterized. We

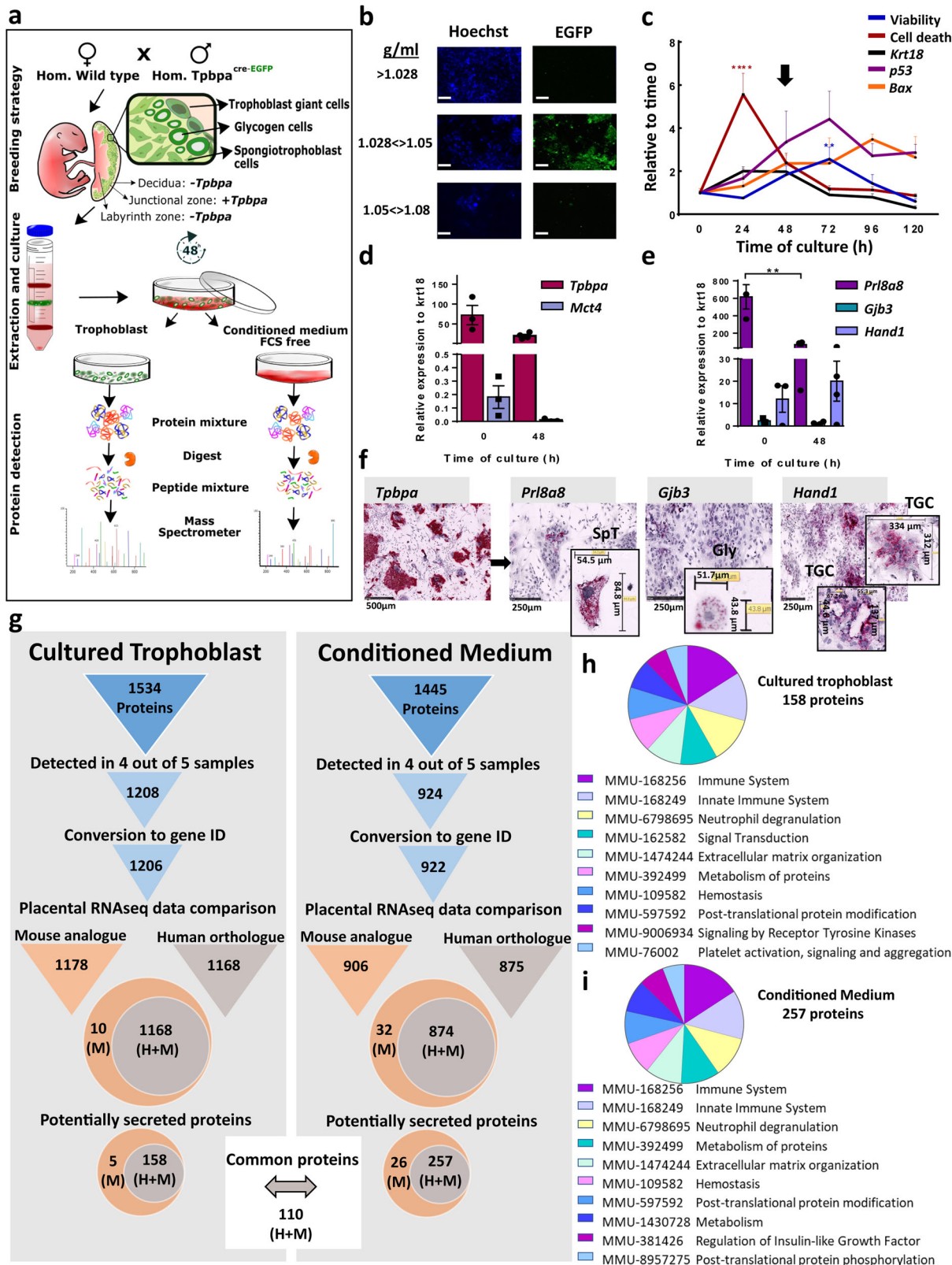

harvested placentas at day 16 of gestation from wild-type females mated with males expressing Cre-EGFP under the *Tpbpa* promoter, which is specifically active in the trophoblast endocrine cells of the junctional zone[13] (Fig. 1a). Day 16 of gestation was chosen as this corresponds to when all the endocrine cells in the mouse placenta have differentiated and are non-proliferating. Moreover, this is when the junctional zone is largest in absolute

terms. The *Tpbpa*-Cre-EGFP reporter was used to visualize the trophoblast endocrine cells, which were found to be enriched in the second layer of our Percoll gradient (between 1.028 and 1.050 g/ml; Fig. 1a, b). Trophoblast cells from this layer were then cultured for up to 120 h and the optimal time point for secretome analysis was identified to be 48 h based on dynamics of trophoblast density (*Krt18* expression) and the levels of viability (XTT

**Fig. 1 Detection of secreted proteins in primary cultures of mouse trophoblast. a** Workflow for the detection of secreted proteins in primary cultures of trophoblast from mice at day 16 of pregnancy. **b** Visualization of EGFP (*Tpbpa*-Cre-EGFP reporter; green) by fluorescence microscopy to identify the Percoll gradient layer containing trophoblast endocrine cells. Cells in layers were counterstained with Hoechst dye (blue) and photographed at magnification ×10 (scale bar 3 mm). **c** Primary cell culture viability (determined by XTT; blue line), cell death (LDH release levels; red line), trophoblast density (*Krt18* gene expression; black line), and apoptosis (*p53* and *Bax* gene expression; purple and orange lines, respectively) from time 0 to 120 h. **d** Proportion of junctional zone (*Tpbpa* gene expression; dark pink bar) and labyrinthine zone (*Mct4* gene expression; light blue bar) trophoblast at 0 and 48 h. **e** Relative abundance of the three junctional zone endocrine cell types, spongiotrophoblast, glycogen cells, and giant cells (gene expression of *Prl8a8, Gjb3,* and *Hand1,* respectively and indicated by purple, green, and light blue bars, respectively) at 0 and 48 h of culture. **f** Representative images of cells stained in situ using RNAscope probes against; *Tpbpa, Prl8a8, Gjb3, Hand1* to visualize trophoblast endocrine cells, spongiotrophoblast (SpT), glycogen cells (Gly), and giant cells (TGC), respectively. **g** Pipeline and results of the analysis of proteins detected by mass spectrometry in cultured trophoblast and their conditioned medium including conversion to RNA sequences and overlay with published RNA data for the mouse and human placenta. Secreted proteins identified using SignalP and combined gene ontology (GO) terms: extracellular region, extracellular exosome, extracellular region parts, and signal IP. **h, i** Pathway over-representation analysis using Reactome pathway by STRING V.11 for the 158 and 257 secreted placental proteins in cultured trophoblast and their conditioned medium, respectively that are expressed by both mouse and human placenta. XTT, LDH, and gene expression data relative to geometric mean of three housekeeping genes: *Gapdh, RPII,* and *Hprt* are presented as mean ± SEM and expressed relative to expression at time 0 h. Asterisks denote statistical significance versus time 0 h, using two-way ANOVA (**b**) or *t*-test (**c, d**), **$p < 0.01$, ****$p < 0.001$, $n = 6$–10 of four pooled litters.

levels), necrosis (LDH levels), and apoptosis markers (*p53* and *Bax* expression) throughout the culturing (Fig. 1c). As expected, the 48 h primary cultures contained a high density of endocrine trophoblasts, as indicated by the high expression of *Tpbpa* and comparatively very low expression of the transport labyrinth zone marker, *Mct4* (Fig. 1d). These cultures contained all three types of junctional zone cells, i.e., the endocrine spongiotrophoblasts, glycogen cells, and giant cells, as evidenced by the expression of their unique gene markers *Prl8a8, Gjb3,* and *Hand1,* respectively (Fig. 1e, f).

**Peptidome and secretome analysis of primary cultured trophoblast cells from mouse placenta.** We then determined the secretome of our primary mouse trophoblast endocrine cell cultures. This involved performing LC-MS/MS on both the cells and conditioned medium from the cultures at 48 h and then applying a bioinformatics pipeline (Fig. 1g). We identified a total of 1534 and 1445 proteins in the cells and conditioned medium of the cultures, respectively. After considering only proteins that were detected in four out of five samples, protein lists were then converted to their corresponding gene ID and expression by the mouse placenta verified using publicly available RNA datasets (Supplementary Table 1). As we wanted to ultimately translate our findings from the mouse to humans, we additionally overlaid our converted mouse gene lists with publicly available RNA datasets for the human placenta (Supplementary Table 1) and performed systematic orthologue searches. To further refine our lists to secreted proteins, we applied SignalP and gene ontology (GO) analysis to capture proteins that employ both the "conventional", as well as "unconventional" secretion pathways (see "Methods" for details). This resulted in a refined list of 158 and 257 secreted proteins detected in the cultured cells and conditioned medium (110 were common between the sample types), respectively, that are expressed by both the mouse and human placenta. Reactome pathway analysis revealed that the proposed functions of secreted proteins in the cultured cells and conditioned medium were largely similar, with the highest scoring pathways including those involved in the immune system, neutrophil degranulation, homeostasis, and IGF regulation (Fig. 1h, i). All data outputs at each step of the pipeline, including the proteins/genes expressed in the mouse but not the human placenta can be found in GitHub (https://github.com/CTR-BFX/2020-Napso_Sferruzzi-Perri).

**Peptidome and secretome analysis of sorted endocrine cells from the mouse placenta.** As a complementary approach to the primary trophoblast cultures, endocrine cells from the mouse placenta on day 16 of pregnancy were isolated using FACS. This was performed by mating the *Tpbpa*-Cre-EGFP mouse line to the double-fluorescent Cre reporter line, mTmG[16] (Fig. 2a). As expected, placentas obtained from these matings showed EGFP in the junctional zone of the placenta, whereas the labyrinth, decidua, and fetus were positive for tdTom (Fig. 2b, c). Moreover, sorting the EGFP-positive cells provided us with highly enriched isolates of junctional zone cells (as indicated by the high expression of *Tpbpa*, with little to low detection of the *Mct4* gene; Fig. 2d) containing the three major endocrine cell types of the mouse placenta (Fig. 2e). Using LC-MS/MS we identified a total of 1142 proteins in the sorted placental endocrine cells (Fig. 2f). Applying a similar pipeline to the analysis of cultured placental endocrine cells, we narrowed down our list of proteins obtained in the sorted placental endocrine cells to 105 secreted proteins that are shared between the mouse and human placenta (Fig. 2f). GO analysis indicated that these secreted placental proteins function in pathways related to the immune system, neutrophil degranulation, and metabolism of proteins, among others (Fig. 2g). All data outputs at each step of the pipeline, including the proteins/genes expressed in the mouse but not the human placenta can be found in GitHub (https://github.com/CTR-BFX/2020-Napso_Sferruzzi-Perri). Proteins detected in the sorted placental endocrine cells that were not predicted to be secreted were analyzed by GO analysis. This revealed that many proteins detected are proposed to play roles in protein synthesis, translation, and metabolism, amongst others and are in line with their possible regulatory role in modulating the function of placental endocrine cells (Supplementary Fig. 1 and Supplementary Data 1).

**Creating a placental secretome map.** Given that the LC-MS/MS method is unbiased but cannot exhaustively characterize the entire proteome of a given set of samples, we then combined the lists of secreted placental proteins expressed by mouse and human placenta obtained using the two methods presented above, to generate a more comprehensive placental secretome map (Fig. 3a). This approach resulted in a total of 319 secreted proteins that are expressed by both mouse and human placenta (Fig. 3b) and another 31 that are specific to the mouse (Supplementary Fig. 2). We aligned our list of 319 secreted placental proteins with data from single-cell RNA-Seq analysis of the human placenta at 8 and 24 weeks of gestation[17] and found that 94% of our proteins (299 out of 319) were expressed in the STB (Fig. 3b, e). We also aligned our list of 319 secreted placental proteins with data on the conditioned media from trophoblast

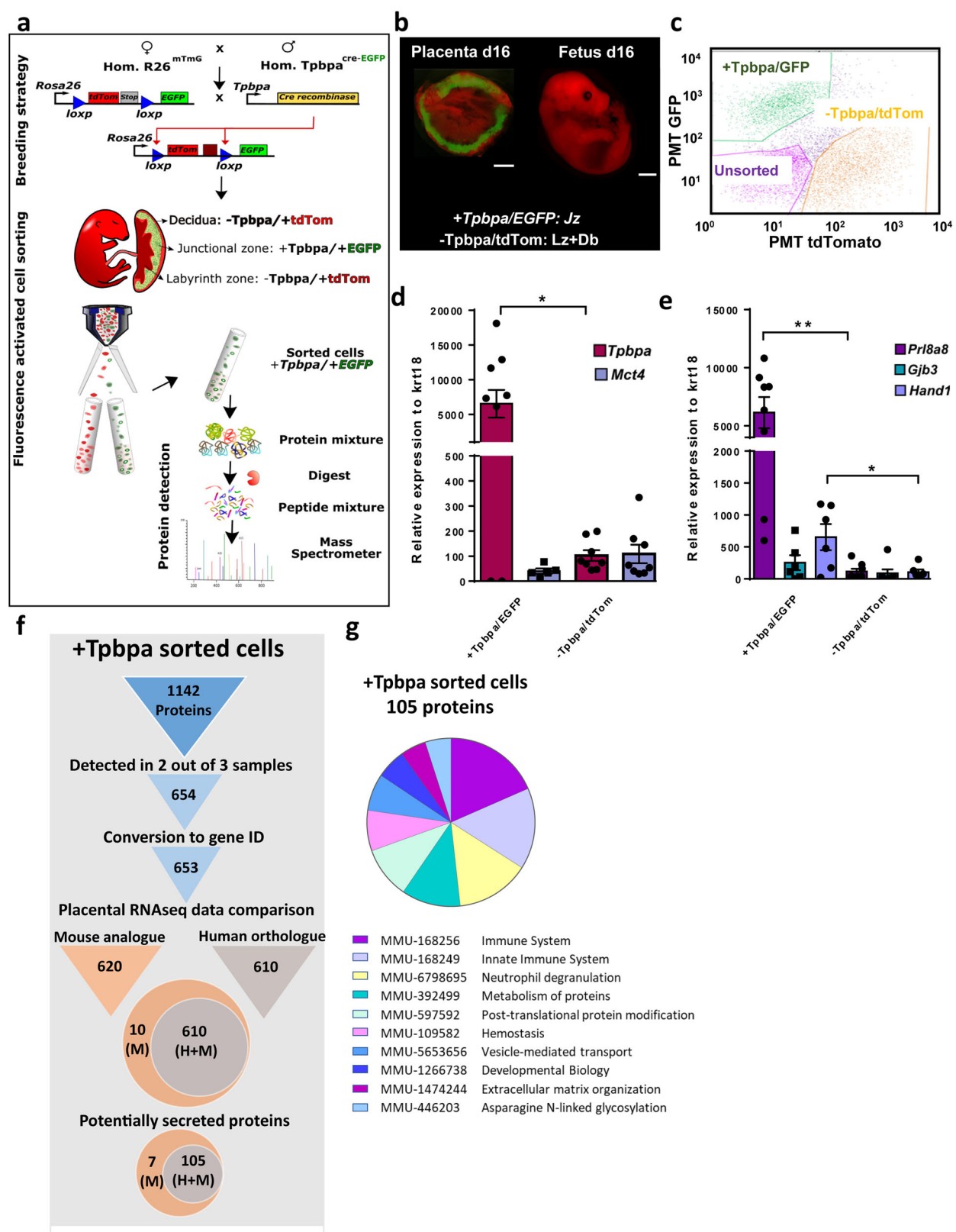

organoids prepared from first trimester human placenta[18] and identified 56 secreted placental proteins in common (Fig. 3b). Notably, 24 proteins of our 319 placental secretome list have been previously identified as potential damage-associated molecular pattern (DAMP) proteins[19]. However, we also found evidence that at least 19 of these are also detectable in mouse plasma, either by us or in previous studies[20,21] (Supplementary Table 2). GO

analysis of the complete list of 319 proteins/genes demonstrated that they play roles in the response to stimuli and stress and regulation of organismal process (Supplementary Data 2). Moreover, many placental proteins identified are implicated in protein and signaling receptor binding and contain protein-interacting domains, such as serpin, conserved sites, and the EGF-like domains, observations which are overall consistent with the

**Fig. 2 Detection of secreted proteins in sorted mouse trophoblast endocrine cells. a** Workflow for the cell sorting and protein expression analysis of mouse trophoblast endocrine cells from mice at day 16 of pregnancy. **b** Fluorescent image of placenta showing EGFP in *Tpbpa* positive cells (junctional zone of the placenta; green) and tdTom (red) for *Tpbpa* negative cells. The fetus for the placenta is also shown, and as expected is only tdTom positive. Scale bar is 250 mm. **c** Representative image of cell sorting of EGFP/tdTom cells by fluorescence-activated cell sorting. **d** Expression of junctional zone and labyrinth zone markers, *Tpbpa* (dark pink bar) and *Mct4* (light blue bar), respectively in the EGFP and TdTom sorted cells. **e** Expression of markers for junctional zone cell types, spongiotrophoblast cell (*Prl8a8*; purple bar), glycogen cells (*Gjb3*; green bar), and giant cells (*Hand1*: light blue bar). **f** Pipeline and results of the analysis of proteins detected by mass spectrometry in sorted *Tpbpa*+/EGFP cells. **g** Pathway over-representation analysis using Reactome pathway by STRING V.11 for the 105 secreted placental proteins expressed by both mouse and human placenta. Data presented as mean ± SEM and genes expressed relative to geometric mean of two housekeeping genes: *RPII* and *Hprt*. Asterisks denote statistical significance to the *Tpbpa*+/EGFP sorted cells, using *t*-test, *$p < 0.05$, **$p < 0.01$, $n = 5$ for each group.

notion that they are secreted (Supplementary Data 2). Using gene expression enrichment analysis for mouse and human tissues, we found that 20 of the proteins were highly expressed (>10 fold) in the mouse placenta compared to other tissues (Fig. 3c) and four secreted placental proteins, TFPI2, SERPINE2, IGF2, and FLT1 were enriched in the human placenta compared to other tissues (Fig. 3d). Further alignment of our complete list of secreted placental proteins with single-cell RNA-Seq analysis[17] of the human placenta revealed that several proteins were enriched predominantly in the STB, including FLT1, TFPI2, and ANGPT2 (Fig. 3e). Moreover, all the proteins that we identified are reported to be expressed by the STB, extravillous cytotrophoblast (EVT), or cytotrophoblast of the human placenta. In a screen of mouse plasma, we found that 45 of the proteins in our placental secretome map were detectable, of which 24 showed a higher relative abundance in pregnancy (between 1.5-fold and 52-fold increase compared to non-pregnant state; Fig. 3f and Supplementary Data 3). A further seven secreted proteins exclusively expressed by the mouse placenta were identified and highly enriched in the plasma of mice during pregnancy, as expected (Supplementary Fig. 3 and Supplementary Data 4).

**Placental secretome map is enriched in proteins that are differentially expressed in human pregnancy complications**. We wanted to know whether our placental secretome map could help us to identify placental proteins that may serve as circulating biomarkers/diagnostic indicators of maternal and fetal wellbeing in human pregnancy. We collated publicly available RNA and protein expression datasets for the human placenta from pregnancies complicated by PE, GDM, IUGR, small for gestational age (SGA), and large of gestational age (LGA) (Supplementary Table 3). We then overlaid our placental secretome map to our collated database of placental RNA/protein expression for these pregnancy complications. This identified 119 secreted proteins that were dysregulated in the human placenta in the pregnancy complications studies (Fig. 4a). There was some overlap in the expression of secreted placental proteins between pregnancy complications and, aside from LGA, all complications showed an altered expression of ANGPT2, FLT1, IGF2, and TIMP2 (Fig. 4a). Of note, FLT1 and IGF2 are particularly enriched in the human placenta compared to other tissues (see also Fig. 3f). Furthermore, we found several secreted placental proteins that were uniquely altered in specific pregnancy complications. For instance, we found 18 secreted placental proteins that were only altered in the placenta of women with GDM, 47 specifically altered in PE and 5 uniquely altered in IUGR pregnancies (Fig. 4a). Of those uniquely altered in PE and IUGR, secreted placental proteins TFPI2 and SERPINE2, respectively, are reported to be highly enriched in the human placenta compared to other tissues. GO analysis revealed that secreted placental proteins uniquely altered in GDM are involved in the metabolism of proteins and extracellular matrix organization (Supplementary Data 5), those altered in PE largely function in the immune system and platelets (Supplementary

Data 6) and those changed in IUGR are implicated in fibril, collagen, and laminin formation (Supplementary Data 7).

**MIF/sFLT1 and ANGPT2/MIF ratios are altered in human GDM blood samples**. We wanted to know whether our secretome map may be useful in identifying placental biomarkers that could be measured in the circulation of women and aid in the detection of a pregnancy complication. To test this possibility, we analyzed the abundance of secreted placental proteins in blood taken from women at booking (12 weeks of gestation) and after glucose tolerance testing (28 weeks of gestation) who were subsequently classified as normoglycemic or diagnosed with GDM. Maternal clinical characteristics and pregnancy outcomes for the women with normal glucose tolerance or GDM are shown in Table 1. We quantified the abundance of the following secreted placental proteins, sFLT1, ANGPT2, MIF, and IGF2 as they were highly enriched in human placenta and/or differentially altered in several pregnancy complications (Fig. 4a). We first visualized the cell-specific expression of these proteins at the maternal–fetal interface in early human pregnancy using the CellxGene tool (https://maternal-fetal-interface.cellgeni.sanger.ac.uk/)[22]. FLT1 was shown to be highly and mainly expressed in STB and EVTs (Fig. 4b), whilst ANGPT2, MIF, and IGF2 were more broadly expressed by trophoblast cell populations (Supplementary Fig. 4). All four secreted placental proteins were detectable in the maternal circulation as early as 12 weeks of gestation (Fig. 4c and Supplementary Fig. 5). Furthermore, several showed changes in abundance with gestational age and in those women, who developed GDM. ANGPT2 and MIF declined in the maternal circulation between 12 and 28 weeks of gestation, in line with the reduction in placental expression indicating that the placenta is the main source (Fig. 3f). However, the decline in MIF with gestational age was not observed in women who developed GDM (due, in part to non-significantly lower values in GDM versus healthy women at 12 weeks). Moreover, sFLT1 circulating levels were overall, significantly elevated in the circulation of women with GDM diagnosis ($p = 0.05$). In contrast, IGF2 levels in the maternal circulation were not significantly different between 12 and 28 weeks of gestation or in women who developed GDM (Fig. 4c and Supplementary Fig. 5). As pregnancy complications can be caused by an alteration of several pathways and biological systems, it is common to evaluate the relationship between the abundance of different biomarkers. We found that the ratio of sFLT1 to MIF concentration was increased by 210% ($p = 0.0003$) and ANGPT2 to MIF was increased by 97% ($p = 0.02$) in women at 12 weeks of gestation who went on to develop GDM compared to the healthy pregnancies (Fig. 4d).

**A transcriptional network controlling placental proteome highlights links with pregnancy complications**. We wanted to gain further information on the regulation of placental endocrine function and its significance for determining pregnancy outcome.

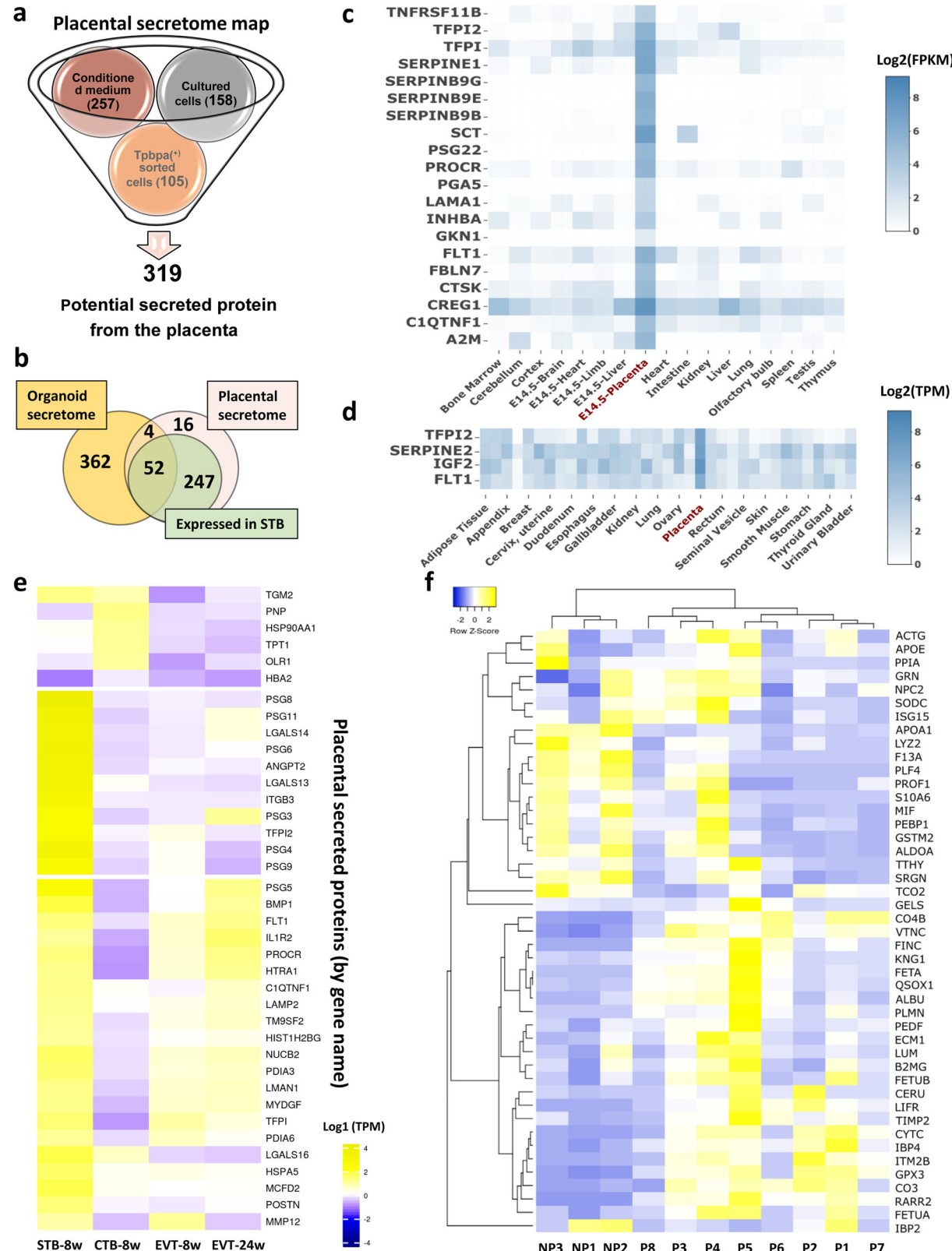

To this aim, we searched for TFs that had significant enrichment of binding sites at the promoters of genes encoding the 319 proteins in our placental secretome map (Fig. 3a). We used two computational tools, Analysis of Motif Enrichment and Ingenuity Pathway Analysis (IPA) and identified 33 common TFs expressed by trophoblast cells in human placenta and controlling the expression of a total of 96 proteins from our placental secretome map (Supplementary

Data 8). All of these TFs were expressed by the STB at 8 weeks of gestation, when their placental secreted targets were too. Furthermore, the expression of ten of these TFs has been reported to be perturbed in human pregnancy complications (Fig. 5a). These ten TFs control the expression of 36 members of the placental proteome map, 20 of which are reported to be differentially expressed in human pregnancy complications (Fig. 5b). Of note was ARNT2,

**Fig. 3 Secretome map of the placenta. a** Integrating the lists of secreted placental proteins expressed by mouse and human placenta and obtained using our different methods to generate a comprehensive placental secretome map. Proteins expressed by mouse but not human placenta shown in Supplementary Fig. 2. **b** Venn diagram showing the overlay of placental secretome map (pink circle) with first trimester trophoblast organoid secretome (Turco[18]; orange circle) and single-cell RNA-seq analysis for human placenta at 8 weeks of gestation (Liu[17]; green circle). Proteins in the placental secretome map that are highly enriched in the placenta of mouse (**c**) and human (**d**) compared to other tissues using TissueEnrich and indicated by the darkest of blue color. **e** Cell-specific expression of the top 30 most highly expressed genes in the placental secretome map using single-cell RNA-seq data for the human placenta (see "Results" text for details). Data represent results after adjusting $p$ values based on Bonferroni correction with highest log1 expression indicated by yellow and lowest log1 expression indicated by dark blue. **f** Heat map generated using Heatmapper showing the relative abundance of peptides secreted by the placenta (MS peak areas normalized to the bovine insulin internal standard), which are detected in non-pregnant ($n = 3$) and pregnant mouse plasma (on day 16 of pregnancy, $n = 8$) with highest row Z-score indicated by yellow and lowest row Z-score indicated by dark blue. CTB cytotrophoblasts, EVT extravillous trophoblasts, NP non-pregnant, P pregnant, STB syncytiotrophoblast, w weeks.

which is reported to be dysregulated in PE and IUGR and implicated in the control of ten genes encoding proteins in the placental secretome, of which five were further reported to be altered in pregnancy complications (Supplementary Data 8). PLAG1 and CREB1 were specifically altered in the placenta from women with GDM and proposed to be key in regulating of placental secreted proteins like IGF2 and FLT1, which were also identified as altered in expression in such pregnancies. FOS, MYCN, and NFYC were found to be altered in PE pregnancies and are predicted to modulate the gene expression of up to 11 proteins in our secretome list that are also reported to be differentially expressed in PE (Fig. 5a and Supplementary Data 8).

## Discussion

Our study has established a comprehensive secretome map of the placenta. By utilizing transgenic mouse lines for tracking placental endocrine cells, together with advanced molecular techniques and bioinformatics analysis, we have characterized a placental secretome map relevant for both mouse and human pregnancy. To achieve this, we performed mass spectrometry on three types of samples: (1) primary cultures containing mouse placental endocrine cells, (2) conditioned media from primary cultures of mouse placental endocrine cells, and (3) endocrine cells isolated from the mouse placenta by FACS. A robust bioinformatics pipeline was then used to integrate our proteins lists and to include in our analysis only proteins expressed by both the mouse and human placenta, as well as those destined to be secreted. By overlaying our secretome map to publicly available datasets for the human placenta, we were able to identify that several secreted placental proteins were altered in pregnancy complications including GDM, PE, and IUGR. Moreover, in proof-of-concept experiments using blood samples from women who were healthy or developed GDM, our findings suggest that the relative abundance of secreted placental candidates identified using our secretome map, may be altered as early as week 12 of gestation, which predates traditional clinical diagnosis at 24–28 weeks. Lastly, we identified TFs that are likely to govern the expression of placental hormones with important implications for pregnancy outcome. Taken together, our data demonstrate that our methodology and placental secretome map may illuminate promising biomarker candidates as early diagnostic indicators and therapeutic targets for pregnancy complications linked to placental malfunction. Moreover, our methodology and findings may have relevance for understanding the significance of placental endocrine function in mammalian development and pregnancy physiology, more broadly.

Three parallel approaches were used to obtain lists of secreted placental proteins that could be integrated as a secretome map. This was fundamentally important, as we wanted to maximize our ability to detect secreted placental proteins without being limited by sample preparation, method sensitivity and specificity. We adapted published methods[23] and used the *Tpbpa*-Cre-EGFP

mouse line[13] to obtain primary cell cultures containing a high density of mouse placental endocrine cells. By monitoring the behavior of our cultured cells, we were able to show that at 48 h of culture, cell viability was stable, with no increase in cell necrosis or cell death. Furthermore, at 48 h, our primary cell cultures contained the three main endocrine cell types of the mature mouse placenta. We analyzed the cultured cells and conditioned media separately, as proteins can be secreted at low concentrations into the culture media, resulting in recovery difficulties. Moreover, salts and other compounds in the media may interfere with protein detection. Furthermore, highly abundant proteins can mask the detection of lowly expressed proteins, resulting in selective detection or a misrepresentation of the proteome when analyzing cultured cells. To further increase the sensitivity of detecting secreted placental proteins, we concentrated the conditioned media of our primary cell cultures prior to mass spectrometry and likely due to this, we obtained a larger list of secreted placental proteins from the conditioned media (26 proteins in the mouse and 257 in mouse and human) compared to the cultured cells (5 proteins in mouse and 158 in mouse and human). However, our primary placental endocrine cell cultures were not pure, and cells may alter their protein expression when cultured. As a complementary approach, we also identified the proteins in freshly isolated mouse placental endocrine cells using the *Tpbpa*-Cre-EGFP and mTmG murine lines and FACS. This approach delivered highly pure samples containing all three placental endocrine lineages. However, some trophoblast giant cells in the mouse placenta may have been lost due to size limitation of the nozzle for the FACS (100 μm). Moreover, both the preparation of primary cultures and fluorescence-activated sorting of mouse placental endocrine cells resulted in a relatively lower abundance of glycogen cells than expected for the mouse placenta at day 16 (normally the relative proportion of endocrine cell types is spongiotrophoblast > glycogen cells > trophoblast giant cells). This is somewhat expected, as glycogen cells are sensitive to lysis and may thus be particularly sensitive to the sample preparation technique. Nevertheless, 112 secreted placental proteins (7 proteins in mouse and 105 in mouse and human) were detected in the pure placental endocrine cell isolates, of which 82% were also found in cultured cells. Reactome pathway analysis of individual protein lists revealed that the proposed functions of secreted proteins in the cultured cells, conditioned culture media, and placental endocrine cell isolates were overall similar, with the highest scoring pathways including those related to the immune system, homeostasis, and IGF regulation. However, several of the proteins detected were specific to one approach/sample type, which reinforces our approach of combining different sample types/methodologies to optimize and broaden the detection of proteins secreted by the placenta. Indeed, the combination of the three methodologies enabled the creation of map of 319 secreted proteins expressed by both mouse and human placenta and another 31 specific to the mouse.

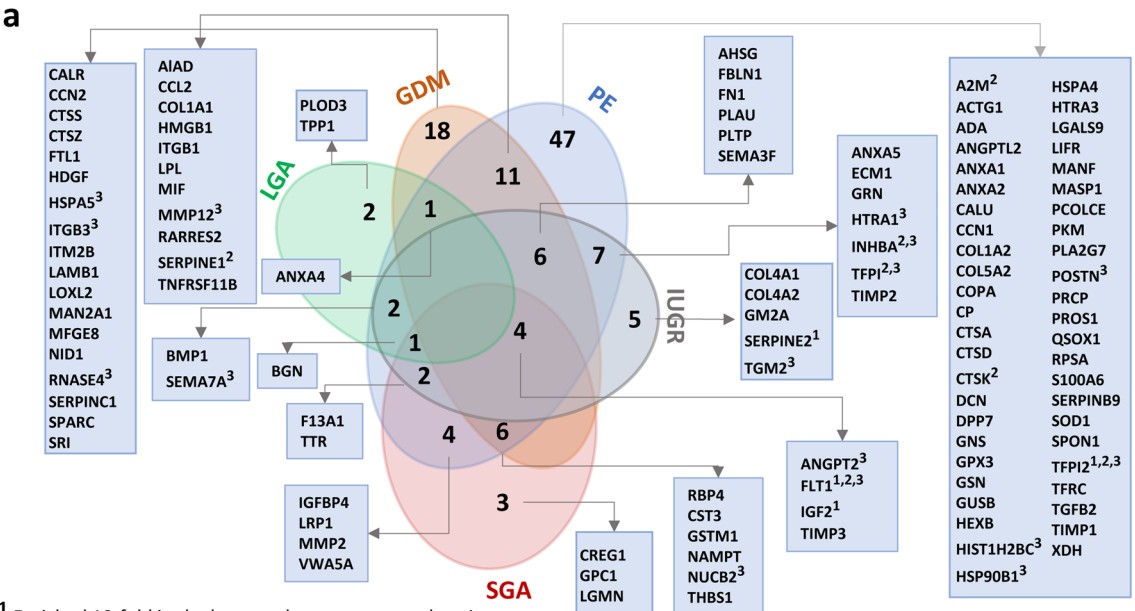

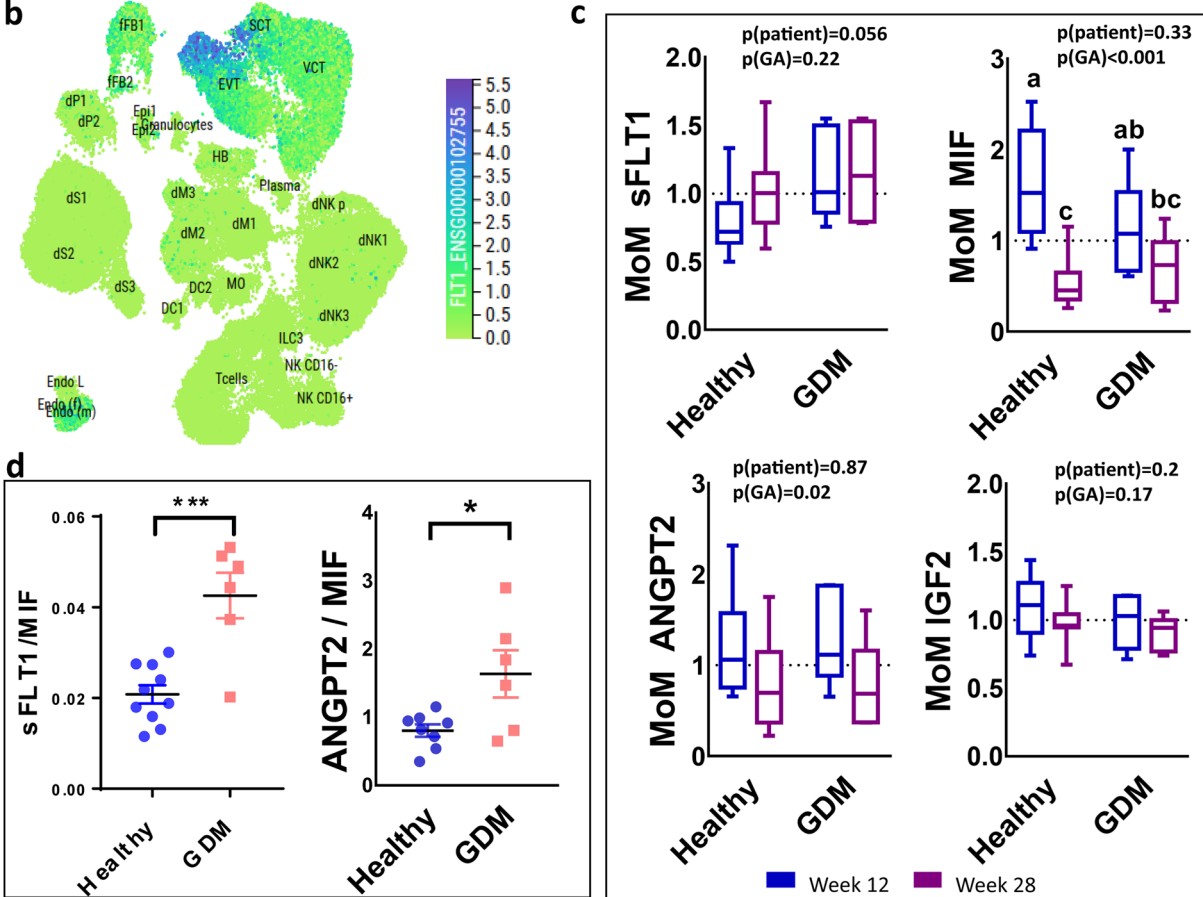

[1] Enriched 10-fold in the human placenta versus other tissues.
[2] Enriched 10-fold in the mouse placenta versus other tissues.
[3] Highly expressed in human Syncytiotrophoblast.

Of the 319 secreted placental proteins in mouse and human, 56 were previously reported to be secreted from first trimester human trophoblast organoids[18], including progranulin (GRN), IGF2, insulin-like growth factor-binding protein 2 (IBP2), and macrophage migration inhibitory factor (MIF). Our analysis also identified a small subset of proteins (24/319) that were previously classified as DAMPs[19]. The fact that we and others also detected

at least 19 of these putative DAMPs in mouse plasma suggests that even if these proteins are the outcome of cellular damage, they could play a physiological role in either cell–cell communication or placental–maternal communication. We validated a subset 45 proteins from our placental secretome map that were detectable in mouse plasma of which more than 50% (24/45) showed a higher relative abundance in pregnancy. Interestingly,

**Fig. 4 Applying the secretome map of the placenta to data available for human pregnancy complications. a** Venn diagram showing the number of placental proteins in the secretome map that are differentially expressed in the placenta of women with a pregnancy complication (LGA: green, GDM: orange, PE: blue, IUGR: gray, and SGA: red ovals. **b** FLT1 expression at the maternal–fetal interface of early human pregnancy via the CellxGene tool (https://maternal-fetal-interface.cellgeni.sanger.ac.uk/). Highest score in blue and 0.0 score in light green. **c** Serum levels of FLT1, MIF, ANGP2, and IGF2 in pregnant women at 12 (blue box plot) and 28 weeks (purple box plot) of gestation who went on to have normoglycemic (healthy) pregnancies or developed GDM. Data are expressed as multiple for the median, MoM. The box blot shows the minimum and maximum values as whiskers, the median line as a solid line and the first (25th percentile) and third quartiles (75th percentile) represented by box length. Raw data for proteins levels are shown in Supplementary Fig. 5. **d** Ratio of the levels of FLT1 to MIF and ANGPT2 to MIF at week 12 of pregnancy in healthy women (blue circles) and those who went on to develop GDM (orange squares). Data are presented as mean ± SEM of MoM. Asterisks denote statistical significance to the GDM pregnancies, using two-way ANOVA (**c**) and $t$-test (**d**), $*p < 0.05$, $**p < 0.01$, $n = 10$–11 for healthy pregnancies, $n = 6$ for GDM. GA gestational age, GDM gestational diabetes mellitus, IUGR intrauterine growth restriction, LGA large for gestational age, PE preeclampsia, SGA small for gestational age.

**Table 1 Clinical characteristics of women used for the analysis of circulating placental hormone abundance in human pregnancy.**

| Characteristics | Healthy pregnancy ($n = 10$) | GDM pregnancy ($n = 6$) | $p$ value |
|---|---|---|---|
| Parity | 1.14 ± 0.26 | 1.29 ± 0.52 | 0.78 |
| Early pregnancy BMI | 33.45 ± 1.78 | 35.37 ± 2.42 | 0.52 |
| GA at OGTT | 28.513 ± 0.75 | 26.55 ± 1.83 | 0.28 |
| OGTT 0 h (mmol/l) | 4.46 ± 0.07 | 5.07 ± 0.2 | 0.002 |
| OGTT 2 h (mmol/l) | 5.65 ± 0.23 | 7.66 ± 0.94 | 0.014 |
| Systolic BP (mm Hg) 1st trimester | 116.8 ± 0.8 | 112.5 ± 1.8 | 0.386 |
| Diastolic BP (mm Hg) 1st trimester | 71.09 ± 0.84 | 68.0 ± 2.1 | 0.576 |
| Systolic BP (mm Hg) 2nd trimester | 117 ± 1.75 | 122.5 ± 4.26 | 0.541 |
| Diastolic BP (mm Hg) 2nd trimester | 70.66 ± 1.6 | 71.75 ± 3.43 | 0.855 |
| HBA1C (mmol/mol) | 33.42 ± 0.58 | 36.0 ± 0.53 | 0.01 |
| GA at delivery | 39.47 ± 0.41 | 38.76 ± 0.57 | 0.30 |
| Birthweight (g) | 3556.36 ± 126.24 | 3392.1 ± 170.00 | 0.44 |
| Infant sex (F/M) | 5/5 | 2/4 | |

Data are shown as mean ± SEM and analyzed by $t$-test.
GA gestational age, BP blood pressure, F female, M male, OGTT oral glucose tolerance test.

several prolactins were also detectable in the plasma of pregnant mice, which is remarkable given that 99.9% of proteins in the circulation are comprised of ~20 high abundant plasma proteins, severely limiting the ability to detect low abundant proteins. However, given that the mass spectrometry analyses are not capable of exhaustively detecting all proteins, we cannot exclude lack of pregnancy-related changes for any of the other 319 proteins in our placental secretome map in the circulation and hence further verification is needed in the future. Indeed, the majority of the proteins in our secretome map (300 out of 319) were localized to the STB of the human placenta[17], which is in direct contact with maternal blood and the primary site for hormone production in the human placenta. The predominate localization of proteins in our secretome map to the STB in humans validates our method and highlights how data generated may improve our understanding of the role and regulation of human placental endocrine function. More than 60% of the proteins in our placental secretome map were predicted to function in "response to stimulus". This GO term includes secondary biological groups such as regulation of signal transduction downstream of the interleukin-1 receptor type 2 (IL1R2), macrophage metalloelastase (MMP12), and apolipoprotein A-I (APOA1) that participate in the response to cytokines. It also included insulin-like growth factor-binding proteins (IGFBP2, IGFBP4, and IGFBP6), which modulate the mitogenic and metabolic actions of IGF that play important roles in pregnancy physiology[24]. Indeed, several of these proteins have been previously shown to be secreted from the human placenta, such as the IL1R2[25], IGF2 and IGFBPs[26], MMP12[27], and APOA1[28] and are consistent with our findings. Many of the proteins in our secretome map were also identified to be hormone-binding proteins, or proteins that regulate signaling downstream of receptors. These included annexin A5 (ANXA5)

an anticoagulant protein, inhibin beta A chain, which is a subunit of both inhibin and activin, transthyretin (TTHY) a thyroid hormone-binding protein, insulin-degrading enzyme (IDE), which binds to insulin and leukemia inhibitory factor receptor (LIFR). Whilst proteins like ANXA5, inhibins/activins, TTHY, and LIFR have been previously reported to be secreted from the placenta[1,29–31], we are not aware of any studies describing the secretion of other proteins, like IDE from the human placenta. Furthermore, other secreted proteins such as adipocyte enhancer-binding protein 1 (AEBP1) and Y-box-binding protein 1 (YBOX1), which can regulate transcription were also detected in the placental secretome. Again however, to date, there are no studies related to the secretion of AEBP1 and YBOX1 from the human placenta. Collectively our findings may suggest that the secretome map comprises known and novel secreted placental proteins.

Two main protein domains featured in our placental secretome map were the "Serpin conserved site" and the "EGF-like domain" which are important in regulating inflammatory processes, growth factor signaling and extracellular matrix and cytoskeletal remodeling. Thus, in addition to identifying proteins secreted by the placenta with systemic actions, proteins in our secretome map may also play local autocrine and paracrine roles in modulating processes at the fetal–maternal interface, including decidual remodeling/function, immune tolerance, and placentation. Using tissue gene expression enrichment analysis, we found that of the proteins in our placental secretome map, 20 were most abundantly expressed by the placenta in mouse and 4 were enriched in the placenta in human. Of note, TFPI2 a plasmin-mediated matrix remodeling, FLT1 a receptor for vascular endothelial growth factor and proteins from the SERPIN family were enriched in both the mouse and human placenta. This is consistent

**a**

| TF | Motif | IPA (p value) | AME (adjusted p value) | Common (DEG in pathology complications) | Complications (TF as DEG) | Reference |
|---|---|---|---|---|---|---|
| ARNT2 | | 14/319 (1.71E-06) | 174/319 (1.30E-13) | 10 (5) | PE SGA | Leavey et al. 2015 Sober et al. 2015 |
| ELF3 | | 3/319 (1.04E-02) | 161/319 (5.07E-08) | 2 (2) | PE IUGR | Sober et al. 2015 |
| PLAG1 | | 3/319 (4.34E-02) | 185/319 (1.82E-04) | 2 (1) | GDM | Sober et al. 2015 |
| SP2 | | 2/319 (1.87E-02) | 166/319 (2.45E-51) | 2 (1) | IUGR LGA PE | Sober et al. 2015 |
| MEF2D | | 6/319 (5.59E-04) | 124/319 (3.84E-23) | 4 (3) | IUGR PE | Sober et al. 2015 |
| MYCN | | 25/319 (5.51E-13) | 71/319 (1.40E-08) | 2 (2) | PE | Leavey et al. 2015 |
| FOS | | 49/319 (4.45E-24) | 41/319 (1.37E-02) | 7 (3) | PE | Gormley et al. 2017 |
| NFYC | | 4/319 (2.93E-03) | 38/319 (7.54E-10) | 2 (1) | PE | Leavey et al. 2015 |
| CREB1 | | 22/319 (2.69E-05) | 10/319 (1.83E-02) | 2 (2) | GDM | Bari et al. 2016 |
| IRF3 | | 9/319 (8.50E-03) | 162/319 (5.83E-54) | 6 (3) | GDM PE | Sober et al. 2015 Leavey et al. 2015 |

**b**

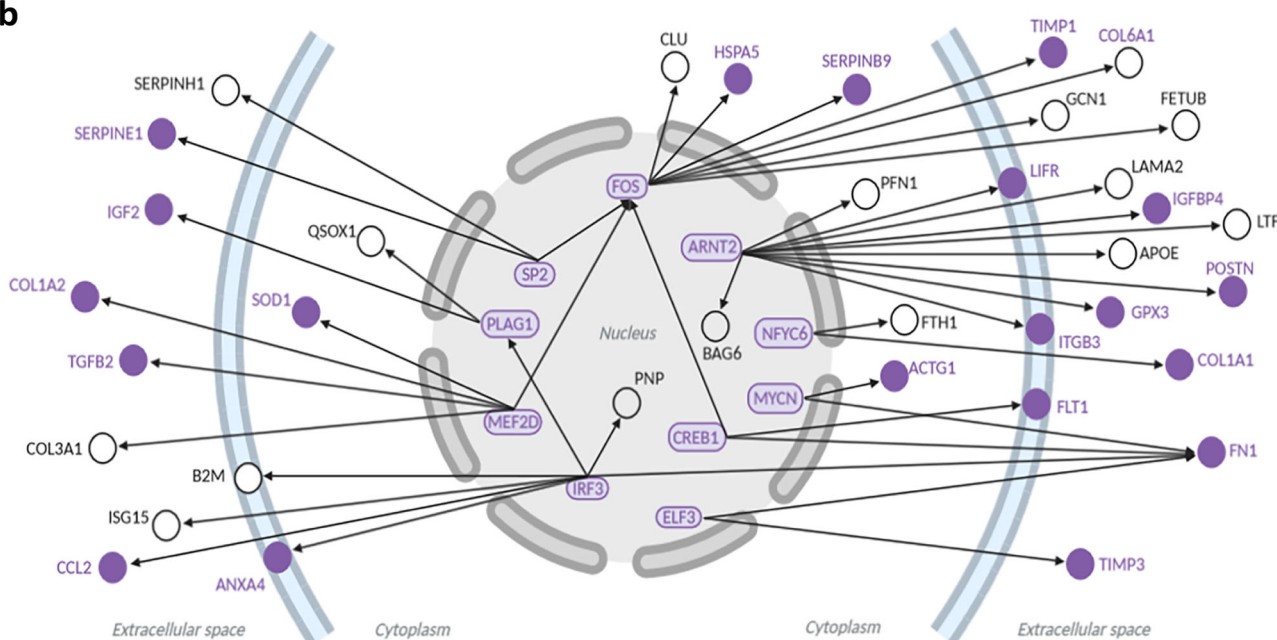

**Fig. 5 Transcription factors dysregulated in pregnancy complications identified as possible regulators of genes encoding secreted placental proteins. a** Table showing the ten transcription factors with altered expression in pregnancy complications and with binding sites enriched at the promoters of genes encoding proteins in the placental secretome map. **b** Regulatory network built with the ten transcription factors listed in **a**. The ten transcription factors and their targets that are differentially expressed in pregnancy complications are highlighted in purple. Location of the target proteins is according to their main cellular/extracellular compartment indicated by IPA. The graph was generated using BioRender.

with the findings of others showing genes and proteins in the placenta overlap in the two species[14,15]. Indeed, as a key aim of our work was to create a secretome map that would be applicable for both human and mouse, we used orthologue searches in our initial steps in narrowing our list of protein candidates. We found that the majority of the proteins we found in our mouse placental endocrine cell protein lists were also expressed by the human placenta, which is higher than reported previously for other placental datasets. However, of note, we found that there were an additional 31 secreted proteins in our complete placental secretome map, which were only expressed by the mouse and not the human placenta. These included 17 members of the PRL/PL gene family and 7 members of the cathepsin gene family, which have both undergone robust species-specific expansion, particularly in rodents, and exhibit unique spatial expression patterns by endocrine trophoblast lineages in mice[32,33]. Overall, PRL/PL and cathepsin family members are thought to play divergent roles in driving physiological changes in the mother during pregnancy, including modulation of insulin production and action, vascular remodeling and immune system regulation. However, further work is required to understand the significance of individual family members to determining pregnancy outcome.

Using a collection of published datasets, we identified that around 30–40% of proteins in our placental secretome map (319 proteins expressed by human and mouse placenta) were reported to be aberrantly expressed by the human placenta in pregnancy complications. Moreover, we found four secreted placental proteins to be differentially expressed in the placenta of women who developed PE, GDM, IUGR, and SGA. This suggests that there may be common pathways related to the production of specific placental hormones, which may underlie or be reflective of the development of such pregnancy complications. Alternatively, the common expression of secreted placental proteins between these four pregnancy complications may reflect that there is overlap between these pregnancy complications, as PE often leads to IUGR or SGA and GDM is linked to PE. Interestingly, two of these proteins; FLT1 and IGF2 were highly enriched in the human placenta compared to other tissues. Moreover, circulating FLT1 (sFLT1) has been previously reported as a suitable biomarker for PE, IUGR, and SGA. Whilst the expression of the FLT1 gene is reported to be altered in the placenta of women with GDM, previous work has not evaluated whether sFLT1 levels are altered and could serve as a biomarker for GDM pregnancies. Similarly, IGF2 was also reported to be altered in the placenta in PE, IUGR, GDM, and SGA pregnancies, however, less is known about changes in circulating IGF2 in the mother[34–39]. Aside from being altered in LGA pregnancies, ANGPT2 and TIMP3 in our secretome map were also reported to be differentially expressed by the placenta in the pregnancy complications assessed. ANGPT2 has previously been explored as predictive candidate biomarker for PE, however it was shown to be unsuccessful[40]. In the context of GDM, ANGPT2 has been reported be differentially secreted from cultured trophoblast of the term placenta in GDM pregnancy[41], but no maternal serum analysis was described. Similarly, altered placental expression of ANGPT2 was reported for SGA[42] and one study has found it to serve as a predictive biomarker for IUGR[43].

We identified several secreted placental proteins altered specifically in some of the pregnancy complications we studied. Analyzing the GO terms of these unique groups of proteins revealed that for those uniquely altered in PE, they were implicated in the control of immune processes, platelet aggregation, and leukocytes like neutrophils, which is in line with previous work focussed on understanding the pathogenesis of PE in women[44]. Similarly, GO analysis of the secreted placental proteins specifically altered in GDM featured terms including protein

metabolism, extracellular matrix remodeling, and the control of the unfolded protein response and is consistent with studies exploring the etiology of GDM[45]. Secreted placental proteins changed in IUGR are proposed to be involved in fibril, collagen, and laminin interactions, however the functional relevance of this finding requires further study.

As a proof of concept that our methodology and placental secretome map may be beneficial for illuminating potential circulating biomarkers with clinical relevance, four secreted placental proteins (sFLT1, MIF, ANGPT2, and IGF2) were assessed in the serum of women who had normal or GDM pregnancies. All four proteins were detectable as early as 12 weeks of gestation. We found that across the two gestational time points, sFLT1 was overall higher in the circulation of women who developed GDM. This was despite the fact that the women with GDM were normotensive. Studies previously exploring changes in sFLT1 in GDM pregnancies have yielded inconsistent results. There was no difference in circulating sFLT1 concentration in lean women with and without GDM in the third trimester of pregnancy[46], and no difference in the secretion of sFLT1 by term placental explant from women who developed GDM[47]. However, sFLT1 has been reported to be elevated in the early second trimester (16–20 weeks of gestation) in women who went on to develop GDM in later pregnancy[48], which is in line with our findings. The relationship between placental sFLT1 production and the development of GDM warrants further study.

We found that the circulating levels of ANGPT2 and IGF2 were not different between women with normal glucose tolerance or GDM at either 12 or 28 weeks of gestation. However, circulating levels of MIF in the mother failed to decline in women with GDM between these two gestational time points. Moreover, the ratios/relationships between the concentrations of MIF to sFLT1 and ANGPT2 to MIF were altered in women at week 12 of pregnancy, which is at least 10 weeks before diagnosis of GDM, suggesting that they may serve as potential early biomarkers for GDM. MIF promotes the secretion of insulin from beta cells and also increases glucose uptake by skeletal muscle[49], thus its significance in development of GDM should be explored further. The value of assessing the ratio/relationship between the concentration of different biomarkers in the maternal circulation for early detection of pregnancy complications is supported by previous work showing a change in the ratio of sFLT1 to PIGF for PE, as well as IUGR in women[2,50]. However, it is important to note that while many previous studies assessing pregnancy biomarkers have used multiple of the median (MoM)[51–53] as we have done so here, the absolute values for the proteins quantified were not statistically significant. In addition, the significance observed with the ratio of MoM values may be due to the calculation placing greater weighting on the MIF results. Finally, the number of women with normal glucose tolerance and GDM in our study was small, and both groups had an elevated body mass index. Further work is required to validate our findings of a change in the ratios/relationships between MIF to sFLT1 and ANGPT2 concentrations in larger cohorts and in women who have a normal body mass index.

It remains to be determined whether more promising data may be obtained from a secretome map of the mouse placenta at an earlier time point in pregnancy, which is more equivalent to the time points examined in our human validation study (day 16 of mouse pregnancy may be considered the start of the last trimester of human pregnancy, which is much later than weeks 12 and 28 of human gestation when screening tests take place). Moreover, work is required to further explore the clinical value of generating a secretome map of the placenta utilizing our approach and if data on multiple secreted proteins or a network of proteins would improve the accuracy of diagnosis of complications like GDM.

Finally, it should be acknowledged that while most hospital laboratories do not have access to sophisticated proteomic analysis, many proteins identified by the current study can be measured in other ways, for example, using immunoassays, which could be automated, facilitating widespread availability. Techniques such as multiplexing also make it feasible to run multiple related assays simultaneously. Indeed, the antenatal screening program (first or second trimester) is a good example of the ability of multiple measured biomarkers to identify high-risk pregnancies, with improved performance using a panel of biomarkers over a single test. While this may not be immediately accessible to hospital laboratories in lower-resource settings, many countries run an antenatal screening program already, and integration of new test panels into that existing infrastructure is feasible. The aim of this study was not to provide new clinically validated biomarker panels ready for diagnostic/predictive use, but rather to provide a resource to support future development of biomarkers or biomarker panels, which may be very valuable in multiple disease areas.

We identified 33 TFs that are expressed by the STB of the human placenta and are predicted to control the gene expression of ~30% of the proteins in our placental secretome map (and are also expressed by the STB at the same time). Of these, ten were previously linked with pregnancy complications. These included ARNT2, which is dysregulated in the placenta of women with PE and IUGR and modulates the expression of genes that are also reported to be altered in human pregnancy complications, such as ANGPT2[12]. ARNT2 is implicated in mediating cellular responses to stimuli including hypoxia[54] and our findings are consistent with recent work applying an integrated systems biology approach[55]. We also identified TFs, PLAG1, and CREB1 that were altered in the placenta of GDM pregnancies and were predicted to modulate the expression of numerous proteins in our secretome map including those that were additionally differentially expressed in GDM, such as IGF2 and FLT1. Consistent with this, CREB1 is regulated by metabolic stimuli like glucose[56] and PLAG1 was pinpointed as a critical factor altered in women who developed a complicated pregnancy[57]. In addition, both PLAG1 and CREB1 have been reported to be dysregulated in mouse genetic models showing placental dysfunction and/or poor pregnancy outcome[58,59]. Thus, numerous TFs likely govern the endocrine function of the placenta and may have significance for understanding the pathogenesis of human pregnancy complications. Hence, future work should center on testing the significance and upstream regulators of TFs identified as putative regulators of placental hormone production.

Many of the biomarkers or functional and transcriptional pathways identified using our approach have been previously suggested or shown to be associated with adverse pregnancy outcomes. This validates our methodological approach, as well as the findings of others that such markers/pathways may indeed be involved in the pathogenesis or serve as diagnostic indicators of those pregnancy complications, thereby reinforcing further work on them. However, it is important to note that several previously unrecognized or understudied markers and pathways were also identified using our approach (e.g., IDE and TTHY as unrecognized placental hormones potentially involved in placental–maternal communication, and unfolded protein response for GDM and neutrophil degranulation for PE), which provides further added value and could be tested more specifically in future work.

In summary, we have generated a comprehensive secretome map of the placenta. This map was proven to be suitable for gaining further information on the significance and regulation of placental endocrine function in mice and humans. Furthermore, we have uncovered different types of secreted placental proteins,

which function in the endocrine and paracrine regulation of maternal physiology, but also possibly in an autocrine manner to modulate placental biology. Sex differences in placental function and pregnancy outcomes are of increasing clinical relevance. As our placental secretome map represents data from both fetal sexes (mouse placentas were obtained from females and males for cell culture and sorting experiments, published datasets used in verification steps did not consider fetal sex, and due to small sample size our human validation study was not split by sex), future work should investigate the extent to which fetal sex modifies the placental secretome and its implication for maternal physiological adaptations during pregnancy. Whether secreted placental proteins may reach the fetal circulation to modulate fetal growth also requires further exploration. However, our placental secretome map revealed that more than 100 proteins may be differentially secreted from the placenta in complicated human pregnancies. Further work is required to extend our findings, including by employing the same methodology and bioinformatics pipeline however, using placental endocrine cells taken from other gestational ages and pregnant mice exposed to environmental conditions, such as maternal obesity, which is a risk factor for complications like PE, GDM, and abnormal birthweight. This will further build knowledge on the role and control of the endocrine placenta during pregnancy and may pave the way for the discovery of novel or improved biomarkers for early detection and prevention of pregnancy complications.

## Methods

**Mouse studies**. Mouse studies were approved by the University of Cambridge Ethical Review Panel and performed in accordance with the UK Home Office regulations Animals (Scientific Procedures) Act 1986 (project licence 70-7645 and PC6CEFE59). All mice used (wild type and transgenic) were on a C57BL/6J genetic background and housed under 12 h dark–light conditions with free access to water and the standard diet (RM3, special dietary services) used in the University of Cambridge Animal Facility. For the preparation of primary cultures of junctional zone trophoblast endocrine cells, wild-type females were mated with males homozygous for *Tpbpa*-Cre-EGFP transgene[13]. For the FACS of junctional zone trophoblast endocrine cells, *Tpbpa*-Cre-EGFP homozygote males were mated to females homozygous for the double-fluorescent Cre reporter construct, mTmG, which expresses membrane-targeted tdTomato prior to, and membrane-targeted EGFP following, Cre-mediated excision (kind gift from Dr. Marika Charalambous, King's College London[16]). The day a copulatory plug was found was denoted as day 1 of pregnancy (term is 20.5 days). Placentas were harvested from mouse dams that were schedule 1 killed by cervical dislocation on day 16 of pregnancy. Female mice were 8–14 weeks when mated and stud male mice were more than 16 weeks of age when used for time-matings.

In a separate set of experiments, age-matched virgin and pregnant (day 16) wild-type C57BL/6J females were anaesthetized with an intraperitoneal injection of fentanyl (sublimaze or fentadon): midazolam (hypnovel): ketamine (ketavet/narketan/ketaset) in sterile water for injection (1:1:0.16:1.63) at 0.15 ml/10 g. After confirming anesthesia, the chest cavity was exposed, and blood was collected by cardiac rupture prior to confirming the animal was schedule 1 killed by cervical dislocation. Blood was collected into an EDTA tube, shaken and placed on ice until it was centrifuged at 3000 rpm for 10 min and plasma recovered and stored at −20 °C until analysis.

**Preparation of primary cultures of junctional zone trophoblast endocrine cells**. Placentas (average of 6–8 per mouse dam) were enzymatically dissociated using a buffer (Medium 199 with Hank's salts, 20 mM HEPES, 10 mM NaHCO, 50 U penicillin/ml, and 50 pg streptomycin/ml, pH 7.4) containing 0.1% collagenase and 0.002% DNase at 37 °C for 1 h, as described previously[23]. Dissociated samples were passed through a 200 μM nylon filter to remove tissue debris and cells were centrifuged at $500 \times g$ for 5 min. Cell pellets were resuspended with Medium 199 ×1 and cells subsequently separated using a three-layer Percoll density gradient (1.028, 1.05, and 1.088 g/ml) according to manufacturer instructions (Percoll plus, GE Healthcare Life Sciences) and as described previously by centrifuging at $600 \times g$ for 30 min with controlled acceleration and braking. Layers from the density gradient were recovered into Medium 199 X1 to dissolve the Percoll solution. Cells were then centrifuged for 5 min at $500 \times g$ and further washed with PBSX1 prior to counting using Haemocytometer. Cells isolated from each layer of the Percoll gradient were fixed with 4% PFA for 20 min and subjected to 5 μl/ml of Hoechst solution for 10 min at 37 °C. Cells were then visualized using fluorescence microscopy (Leica TCS SP8 Confocal laser scanning microscope) and the second layer was found to contain the greatest density of junctional zone trophoblast

endocrine cells (EGFP-positive cells due to Tpbpa-Cre-EGFP) and therefore used for further analysis. Namely, cells in the second Percoll density layer were seeded in 96-well plates, 8 chamber slides or in 6-well plates ($10^5$ cells/ml) and time of seeding was defined as time 0. Cells were grown in NTCT-135 medium containing 10% fetal bovine serum, 50 IU/ml ampicillin, 50 μg/ml streptomycin, 2 mM l-glutamine, 20 mM HEPES, and 10 mM NaHCO and maintained in a humidified atmosphere of 5% $CO_2$ at 37 °C. Cell medium was replaced every 24 h. Cells were washed three times in PBSX1 and serum-free medium applied 24 h prior to any downstream analysis.

**Viability assay**. Cell viability was determined using an XTT cell proliferation assay kit (Abcam—ab232856) according to manufacturer's instructions. Values were calculated as % of values at time 0 h and each experiment was performed in triplicate in 8–10 independent assays.

**Cell death assay (LDH release assay)**. Cell death was determined by measuring the activity of lactate dehydrogenase (LDH) in the conditioned media of primary cell cultures using a LDH cytotoxicity assay kit, according to the manufacturer's instructions (Thermo Scientific). Cell-free medium and cells treated with medium containing Triton-X were used as negative and positive controls, respectively. Each experiment was performed in triplicate in six independent assays.

**In situ hybridization**. In situ hybridization was performed on primary cell cultures seeded on chamber slides at 90% confluency ($10^6$ cells/well) (Thermo Fisher Scientific, UK). Cells were fixed in 4% PFA for 30 min, washed twice with PBSX1, dehydrated in increasing concentrations of ethanol and stored in 100% ethanol at −20 °C. In situ hybridization was performed using the RNAScope 2.5 RED chromogenic assay kit following the manufacturer's instructions (Bio-Techne, UK). Briefly, slides were allowed to equilibrate to room temperature and rehydrated in PBSX1. RNAscope® Hydrogen Peroxide was applied to the slides for 10 min at RT, followed by RNAscope® Protease Plus in RT for 10 min. Slides were then incubated with the target or control probes at 40 °C for 2 h (negative control probe (310043), positive control probe (313911), Tpbpa-probe (405511), Prl8a8-probe (528641), Gjb3-probe (508841), and Hand1-probe (429651) in a HybEZ oven for 2 h at 40 °C. Next, slides were washed twice with wash buffer and were subjected to six rounds of amplification and the probe signal was developed via a reaction with fast red. Slides were then counterstained with Haematoxylin and mounted in EcoMount. Slides were scanned on a NanoZoomer 2.0-RS slide scanner (Hamamatsu, Hamamatsu City, JP) at ×40 magnification.

**Conditioned medium preparation for mass spectrometry**. Conditioned medium from cultured cells was collected at 48 h of culture. At 24 h prior to medium collection, cells were washed three times with PBSX1 and cultured in serum-free medium. The conditioned medium was centrifuged at $1000 \times g$ for 10 min and total protein concentration measured using a Bradford assay. Proteins in the conditioned medium were concentrated up to 1.2 μg/μl of total protein using cellulose membrane Ultra-4 Centrifugal Filter Unit of 3 KDa cut-off (Merck) as per the manufacturer instructions.

**RNA extraction and quantitative real-time PCR (qPCR)**. Total RNA was extracted from cultured cells using RNeasy Plus Mini Kit (QIAGEN) and 0.5 μg of RNA was reverse transcribed with random hexamer primers using a High-Capacity cDNA RT kit (Applied Biosystems) according to the manufacturer's instructions. qPCR was performed using MESA BLUE qPCR Master Mix (Eurogentec) on a Bio-Rad T100 thermocycler using gene-specific intron-flanking primers. In particular, forward and reverse primers for Tpbpa are: TGAAGAGCTGAACCACTGGA and CTTGCAGTTCAGCATCCAAC, Mct4: GGCTGGCGGTAACAGAGTA and CGGCCTCGGACCTGAGTATT, Prl8a8: TCAGAGCTGCATCTCACTGC and GGGACATCTTTCATGGCACT, Gjb3: GGGGCTCTCCTCAGACATA and ACCTGCTAGCCACACTTGCT, Hand1: GGAGACGCACAGAGAGCATT and CACGTCCATCAAGTAGGCGA, and Krt18: CAAGACCTGAACCGTCGCCT and ATTCGCAAAGATCTGAGCCCT, respectively. Gene expression was analyzed in triplicate and the Ct values were normalized to the expression of internal housekeeping genes (Gapdh, RpII, and Hprt). Forward and reverse primers for Gapdh are: GGGAAATGAGAGAGGCCCAG and GAACAGGGAGGAGCAGAGAG, RpII: AGATGTATGACGCCGACGAG and AATCGGTGGTGCATCTTCCA, and Hprt: CAGGCCAGACTTTGTTGGAT and TTGCGCTCATCTTAGGCTTT, respectively. Results are presented as mean ± SEM and relative to time 0 h.

**Fluorescence-activated cell sorting (FACS) of live cells for mass spectrometry**. Single-cell suspensions were prepared from whole placentas digested with the dissociation buffer containing 0.1% collagenase and 0.002% DNase at 37 °C for 1 h, as described for the preparation of primary cell cultures. Cells were filtered through 100 μM nylon filter and centrifuged at $500 \times g$ for 5 min. Cell pellets were resuspended with PBSX1 and 7AAD (Life Technologies, A1310) added prior to sorting using a FACSJazz machine (BD Biosciences, Singapore). Live cells were

gated and cells positive for EGFP or tdTomato were sorted and lysed directly into 80% acetonitrile (ACN) in water (v/v) in a Protein LoBind Eppendorf tube.

**Mass spectrometry**

*LC-MS/MS analysis of conditioned media*. Conditioned media from primary cultures of trophoblast cells were standardized to a final concentration of 2 μg/μl in 4% SDS loading buffer with 100 mM dithiothreitol (DTT). Samples were denatured at 95 °C for 5 min and 10 μg of total protein per sample was loaded onto a 12% SDS PAGE gel and run at 120 Volts. The gel was then stained with colloidal coomassie stain and washed with water. Protein bands in each lane were cut into 1 mm$^2$ pieces, de-stained, reduced (using DTT), and alkylated (using iodoacetamide) and subjected to enzymatic digestion with sequencing grade Trypsin (Promega, Madison, WI, USA) overnight at 37 °C. After digestion, the supernatant was pipetted into a sample vial and loaded onto an autosampler for automated LC-MS/MS analysis. All LC-MS/MS experiments were performed using a Dionex Ultimate 3000 RSLC nanoUPLC (Thermo Fisher Scientific Inc, Waltham, MA, USA) system and a Q Exactive Orbitrap mass spectrometer (Thermo Fisher Scientific Inc, Waltham, MA, USA) as described previously[60].

For medium generated spectra, all MS/MS data were converted to mgf files and the files were then submitted to the Mascot search algorithm (Matrix Science, London UK, version 2.6.0) and searched against the UniProt mouse database (61,295 sequences: 27,622,875 residues) and a common contaminant sequences containing non-specific proteins such as keratins and trypsin (115 sequences, 38,274 residues). Variable modifications of oxidation (M) and deamidation (NQ) were applied, as well as a fixed modification of carbamidomethyl (C). The peptide and fragment mass tolerances were set to 20 ppm and 0.1 Da, respectively. A significance threshold value of $p < 0.05$ and a peptide cut-off score of 20 were also applied. All data (DAT files) were then imported into the Scaffold program (Version 4.5.4, Proteome Software Inc, Portland, OR). LC-MS/MS experiments were partially performed at Cambridge Centre for Proteomics.

*LC-MS/MS analysis of primary cultured cells and sorted cells*. Trophoblast from both primary cell cultures and fluorescence activating cell sorting were treated with 800 μl of 80% ACN in water and centrifuged for 5 min at $10,000 \times g$. The supernatant was removed and the pellet was reduced and alkylated using 50 mM ammonium bicarbonate and 10 mM DTT at 60 °C for 1 h, followed by the addition of 100 mM iodoacetamide in the dark for 30 min. Enzymatic digestion was performed using Trypsin at 10 μg/ml in 50 mM ammonium bicarbonate overnight at 37 °C (enzymatic digestion was halted by the addition of 1% formic acid). Samples were analyzed by LC-MS using a Thermo Fisher Ultimate 3000 nano LC system coupled to a Q Exactive Plus Orbitrap mass spectrometer (Thermo Scientific, San Jose, CA, USA), as described previously[61]. For generated spectra, all LC-MS files were searched using PEAKS 8.5 (Bioinformatics Solutions Inc) software against the Swissprot database (downloaded 26-Oct-2017) with a *Mus musculus* filter. A tryptic digestion approach was selected using a semi-specific setting and up to a maximum of three missed cleavages. The search outputs had a 1% FDR setting applied, along with a unique peptide setting of at least one peptide. The precursor and product ion tolerances were 10 ppm and 0.05 Da, respectively.

*LC-MS/MS analysis of mouse plasma*. Plasma was defrosted and an equal volume of 6 M guanidine hydrochloride was added and the sample mixed thoroughly. Proteins were precipitated by the addition of 75% ACN in water at a ratio of 1:6 (diluted plasma:ACN), and the sample centrifuged at $2900 \times g$ for 10 min at 4 °C. The supernatant was transferred and evaporated under oxygen free nitrogen. The extract was reconstituted into 200 μl of 0.1% formic acid in water and extracted using solid phase extraction. The eluent was evaporated and reduced and alkylated prior to overnight trypsin digestion. Extracts were acidified and analyzed using nano LC-MS and the data searched using PEAKS as described above. Peptides from specific proteins of interest in the PEAKS search result were selected for targeted quantitation in the raw LC-MS files. The most abundant peptide from each protein was selected and their $m/z$ values were used to generate extracted ion chromatograms using the Xcalibur LCQuan program (Thermo Fisher Scientific) and the peak areas of each peptide was expressed as a ratio of a digested peptide from the spiked internal standard peptide (bovine insulin)[61,62].

*Bioinformatics analysis*. Protein/peptide annotations in LC-MS datasets were converted to their gene accession ID via UniProt (https://www.uniprot.org/uploadlists/). Gene lists were then overlaid with publicly available datasets for the mouse and human placenta, which are detailed in Supplementary Table 1 (three from mouse placenta and eight for human placenta). Mouse–human ortholog searches were also undertaken using three sources of data, MGI (http://www.informatics.jax.org/), NCBI (https://www.ncbi.nlm.nih.gov/homologene), and Ensembl (biomaRt_2.42.1 and homologene_1.4.68 in R version 3.6.2; https://www.R-project.org/). Then using R, a combined ortholog list for Mouse–Human was generated (details of the list and Rscript can be found in GitHub, https://github.com/CTR-BFX/2020-Napso_Sferruzzi-Perri). Mouse–human ortholog results were classified as one-to-one when a gene/protein from mouse was found at the end of a node in human. Any results classified as one-to-many were excluded. GO analyses

were performed using both STRING and Panther tools (https://string-db.org/). Gene enrichment analyses were conducted using TissueEnrich (https://tissueenrich.gdcb.iastate.edu/), which utilizes datasets available in the Human Protein Atlas compiling RNA-seq datasets from 35 human tissues and the Mouse ENCODE database comprised of RNA-seq datasets of 17 mouse tissues. Heat map of MS peak areas of each peptide (normalized to the internal standard—bovine insulin) detected in mouse plasma was generated using Heatmapper (http://www.heatmapper.ca/expression/) using Average Linkage as clustering method and Spearman Rank Correlation as distance measurement method, applied to both columns and rows. Refined gene/protein lists were overlaid with publicly available RNA and protein expression datasets for human pregnancy complications (Supplementary Table 3) and aided by searches in Pubmed and the OMIM repository (http://www.ncbi.nlm.nih.gov).

To further refine our lists to secreted proteins, we applied SignalP (Signal Peptide Prediction, version 4.1[63]) and GO analysis using four different GO terms: extracellular region (GO: 0005615), extracellular exosome (GO: 0070062), extracellular region parts (GO: 0005615) and signal IP (excluding signals detected for ER lumen proteins)[64]. This was undertaken because SignalP can only detect the signal peptide for proteins secreted via the canonical route, which is also known as the "classic" or "conventional" secretion pathway. However, eukaryotic cells also utilize an unconventional protein secretion route for protein sorting and delivery, an "unconventional" secretion pathway, including leaderless proteins that are secreted into the extracellular space. This approach allowed us to capture proteins that employ the "conventional", as well as "unconventional" secretion pathways. All data outputs at each step of the pipeline, including the proteins/genes expressed in the mouse but not the human placenta and the refinement of our list to secreted proteins can be found in GitHub (https://github.com/CTR-BFX/2020-Napso_Sferruzzi-Perri). The mass spectrometry proteomics data have also been deposited to the ProteomeXchange Consortium via the PRIDE partner repository with the dataset identifier PXD025006.

To search for enrichment of TF binding motifs at the promoters of the genes encoding the 319 proteins that are part of placental secretome, we first used Eukaryotic Promoter Database (https://epd.vital-it.ch/index.php) to retrieve the DNA sequences from 1000 bp upstream to 100 bp downstream of the transcriptional start site. These sequences were then analyzed using Analysis of Motif Enrichment v4.12.0 (http://meme-suite.org/tools/ame) by selecting *Mus musculus* and HOCOMOCO Mouse (v11 FULL) as motif database. An additional search for upstream regulators was performed using the IPA (Qiagen), and only TFs predicted by both tools ($n = 77$) were selected for further analysis. Next, we filtered for TFs with enriched expression in STB cells from human placenta at 8 weeks of gestation that had at least one common target gene encoding one of the 319 placental secretome proteins ($n = 33$). Literature search led to identification of ten of those TFs that were linked with pregnancy complications. Transcriptional network visualization for the ten TFs and the corresponding targets was performed using IPA.

*Human study population and sampling.* The recruitment of human subjects was performed in accordance with relevant guidelines and regulations and approved by the Research Ethics Committee under agreement REC 18/LO/0477 as part of the Observational study of pregnancy hyperglycemia, endocrine causes, lipids, insulin and autoimmunity study (OPHELIA; approved 5 April 2018; Research Registry number 5528[65]; described in detail elsewhere[66]). All participants provided written informed consent prior to participation. Inclusion criteria included (1) singleton pregnancy, (2) no evidence of severe congenital anomaly, and (3) a referral for an OGTT for clinical reasons, according to NICE guidelines (https://www.nice.org.uk/guidance/ng3). Exclusion criteria for this study were (1) multiple pregnancy, (2) severe congenital anomaly on ultrasound, (3) severe anemia on previous blood tests, (4) previous diagnosis of diabetes outside of pregnancy, and (5) medications at the time of the OGTT, which may interfere with the results of the OGTT. Screening for GDM was performed at 24–28 weeks of gestation using a 75 g OGTT and diagnosis of GDM was made in accordance with the IADPSG glycaemic cut-off values (fasting value ≥92 mg/dl (5.1 mmol/l), 1 h post-glucose load ≥180 mg/dl (10 mmol/l), 2 h post-glucose load ≥153 mg/dl (8.5 mmol/l). Venous blood samples were collected from pregnant women in the first trimester (12 weeks of gestation) and second trimester (28 weeks of gestation) at the same time as routine clinical venesection. Plasma was recovered by 2500 rpm for 10 min and samples were stored at −80° C. Blood pressure measurements were taken at various times in pregnancy using a calibrated automatic oscillometric sphygmomanometer (Dinamap machine) and systolic and diastolic blood pressure recorded. For inclusion in this study, a cohort of patients were chosen retrospectively from the OPHELIA pilot study including six women with GDM and a control group of age- and BMI-matched participants with normal glucose tolerance ($n = 10$).

*Human plasma analysis of protein candidates by ELISA.* Candidate secreted placental proteins were quantified in maternal plasma samples from healthy women and those diagnosed with GDM, using commercially available ELISAs for sFLT1 (K15190D, MSD), MIF (K151XJK-1, MSD), ANGPT2 (F21YR-3, MSD), and IGF2 (DG200, R&D) by the Core Biochemical Assay Laboratory, Cambridge and according to the manufacturer's instructions.

**Statistics and reproducibility.** Data for viability, cell death, and qRT-PCR are presented as mean ± SEM. Data were considered normally distributed (Shapiro–Wilk normality test) and analyzed by two-way ANOVAs (with Tukey correction for multiple comparisons) or *t*-tests to determine significant differences. The number ($n$) of samples per group for each analysis and specific statistical test employed are detailed in each figure legend for every dataset. All analyses were performed using GraphPad Prism version 7.00 (GraphPad Software). Any adjustment of *p* values based on Bonferroni correction are stated in the corresponding figure legend. $p < 0.05$ was considered to indicate a statistically significant difference between the groups analyzed.

**Reporting summary.** Further information on research design is available in the Nature Research Reporting Summary linked to this article.

## Data availability
All source data underlying the graphs and tables are available in Supplementary Data files and/or can be found in GitHub (https://github.com/CTR-BFX/2020-Napso_Sferruzzi-Perri) with corresponding DOI 10.5281/zenodo.4642653[67]. The mass spectrometry proteomics data have also been deposited to the ProteomeXchange Consortium via the PRIDE partner repository with the dataset identifier PXD025006. All other data (if any) are available upon reasonable request.

## Code availability
The in-house scripts used for the analysis can be found in the following online repository: https://github.com/CTR-BFX/2020-Napso_Sferruzzi-Perri[67]. The bioinformatics software used are R version 3.6.2 (corresponding package versions are listed in GitHub), SignalP version 4.1, GraphPad Prism version 7.00, and AME version 4.12.0.

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

## Acknowledgements

We would like to thank Dr. Marika Charalambous, King's College London for the mTmG mouse line. We would like to thank the staff in Combined Animal Facility, as well as Dr. Jorge Lopez-Tello and Miss Bethany R.L. Aykroyd, Centre for Trophoblast Research, Department of Physiology, Development and Neuroscience, University of Cambridge for their assistance in the breeding and husbandry of mice. We would like to thank Dr. Laura Kusinski, Wellcome-MRC Institute of Metabolic Science, Addenbrooke's Hospital, Cambridge for locating the samples from pregnant women who volunteered to participate in the OPHELIA study. Finally, we thank the Bioinformatics and Biostatistics (Bio2) Core Facility of the Wellcome-MRC IMS-MRL for the support

provided by facilitating the access to data analysis tools. This work was supported by a Royal Society Dorothy Hodgkin Research Fellowship, Academy of Medical of Sciences Springboard Grant, Isaac Newton Trust Grant and Lister Institute Research Prize grant to ANSP (grant numbers DH130036/RG74249, SBF002/1028/RG88501, RG97390, and RG93692, respectively). T.N. was supported by an EU Marie Skłodowska-Curie Fellowship (PlaEndo/703160) and an Early Career Grant from the Society for Endocrinology. C.L.M. is supported by the Diabetes UK Harry Keen Intermediate Clinical Fellowship (DUK-HKF 17/0005712) and the EFSD-Novo Nordisk Foundation Future Leader's Award (NNF19SA058974). Work in the FR/FMG laboratory was supported by the Wellcome Trust (106262/Z/14/Z, 106263/Z/14/Z), the MRC (MRC_MC_UU_12012/3 and MRC-enhancing UK clinical research grant MR/M009041/1) and the Cambridge Biomedical Research Centre (NIHR-BRC Gastrointestinal Diseases theme).

## Author contributions

T.N. and A.N.S.-P. designed the study. T.N. performed the animal experiments, cell culture, and FACS studies under the supervision of A.N.S.-P. R.G.K., A.L.G., F.M.G., and F.R. implemented mass spectrometry studies. T.N., X.Z., M.I.L., and I.S. conducted bioinformatic analysis under the supervision of R.S.H. and A.N.S.-P. I.S. also performed transcription factor analysis. T.N., A.N.S.-P., and C.L.M. performed the proof-of-concept study using human samples from the OPHELIA study. T.N., X.Z., and A.N.S.-P. interpreted the results. T.N. and A.N.S.-P. wrote the paper. All authors approved the final version of the manuscript.

## Competing interests

The authors declare no competing interests.
