## [Transparent Peer Review File · Communications Biology]

Reviewers' comments:

Reviewer #1 (Remarks to the Author):

The manuscript by Napso et al identifies proteins secreted by the placenta that could potentially be used as disease pathology markers. This work was initiated using two complementary murine approaches to narrow in on proteins secreted by endocrine-type cells in the junctional zone. Cultured trophoblasts, FACS sorting, and conditioned media were used to enrich these cells for LC/MS-based proteomics. Pathway analyses were used to determine signaling networks that were observed in these models and when compared to known proteins found in human placentas (healthy and complicated pregnancies). This list was narrowed to 4 proteins that were subsequently quantified in the circulation of subjects with gestational diabetes. The paper provides a unique and exciting approach to identify candidate proteins. The paper is also relatively straight forward to follow and well-written.

Major Concerns:

- In Figure 1, the authors were careful to quantify cell death and observed a peak at 24 h. However, the proteomics data were conducted on media collected between 24-48 h. This raises concern that subsets of the proteins in the media may reflect ex vivo cell death (such as DAMPs) associated with culturing of primary cells rather than secreted proteins involved in cell-cell communication or maintenance of placental health. Were any of the secreted proteins from Figure 1G (conditioned media) known DAMPs?
- After identifying the secreted proteins from Figures 1 and 2, did the authors evaluate the relationship of the overlapping proteins in the two models with serum from non-pregnant and pregnant mice? This would help to refine the analysis to placenta-secreted/enriched in the circulation.
- There are concerns with the presentation of the data for the 4 proteins circulating in human pregnancies at 12 and 28 weeks between healthy and GDM. When reviewing the raw data in Supplemental Figure 4, there were no statistical differences between healthy and GDM for any of the proteins quantified. There may be a trend for MIF but no p value is provided. The results section does not acknowledge that there were no differences when comparing absolute values. Then, the authors decided to use two variations on data reporting: 1) multiple of the median (MoM) and 2) ratios. While use of MoM in reproductive sciences was initiated to compare AFP values between laboratories, it does not appear to be warranted in this study as all of the samples were presumably run at the same time on the same ELISAs. Further, in Figure 4D, the authors express FLT and ANGPT2 relative to MIF levels and cite the use of ratios for sFlt, etc to justify this expression. In this case, it appears that the MIF values are what is actually driving the differences in ratios. This is not acknowledged. Likewise, this fact greatly impacts the FLT1 data as the enrichment of the two proteins in the circulation differs by a couple of log units and the ratio places greater weighting as a result on the MIF levels. While it is admirable to explore novel ways to express biomarker data, the fact that there are actually no differences in concentrations across the groups (and that this is not acknowledged) is concerning.

Minor Concern:

- As noted by Jackson labs, there is no such thing as a C57BL/6 mouse. More details regarding the substrain(s) used need to be included.

Reviewer #2 (Remarks to the Author):

In the manuscript "Unbiased placental secretome characterization identifies candidates for pregnancy complications", Napso and colleagues sought to characterize the secretory output by the endocrine cells isolated from mouse placentas using different molecular methodologies and bioinformatic analyses, and then determined if these findings may potentially be relevant to human population by investigating a few of the identified secreted placental proteins in women with GDM. There are very few studies that examine the endocrine functions of the placenta and how this affects pregnancy

outcomes or helps predict pregnancy outcomes. Furthermore, the bioinformatic pipelines used by the authors are novel, and the idea of generating a placental secretome map for outcome prediction or diseases diagnosis is also innovative; these methodologies may be applicable to other research areas outside placenta/pregnancy/reproduction. Hence, this work is significant and likely to be of high interest to a wide range of readers/researchers.

The authors present their methods and results clearly, and are thoughtful in addressing some of the potential problems related to cell culture. Overall, the manuscript is well-written. I have only a few questions and minor suggestions to the authors.

1) My biggest concern with this manuscript is that the authors may have performed both mouse studies and the human validation study without considering the impact of fetal sex. Can the authors provide a rationale for why fetal sex is not considered? Will fetal sex have any influences on the placental secretome and will the resulting map generated from male placental endocrine cells vs female placental endocrine cells differ? Can't fetal sex be included as a variable in the bioinformatic analyses?

2) Although the bioinformatic pipeline and the idea of a placental secretome map are novel and exciting, many of the markers (such as FLT1, IGF2) or functional pathways (such as immune system, growth factor regulation) identified and reported in this manuscript have been previously suggested or shown to be associated with adverse pregnancy outcomes. It is unclear what new insights are gained and learned from generating a placental secretome map using such a sophisticated approach. Can the authors please comment about this in their discussion?

3) Related to the second comment above - although it is true that the secretome map may enable the identification of new secreted proteins, their clinical values and diagnostic accuracy are still largely unclear and remain to be thoroughly investigated. In this context, it is unclear if such information from the secretome map is of clinical relevance. Additionally, if data on multiple secreted proteins or a network of proteins are needed to improve the accuracy of diagnosis (as suggested by calculating the protein ratios), the feasibility of doing so in clinical settings - particularly low resource clinical settings - is unclear. Can the authors please comment about this as well in their discussion?

4) The authors provided a rationale for the studied time point (gestational day 16 in mouse pregnancy), which is reasonable. It will also be a good idea to explain this mouse gestational time point in the context of human pregnancy, especially related to the time points examined in their human validation study (ie: week 12 and week 28).

5) If the LCMS quantification method has been published, please include the reference (Lines #612 and 632). If there are any modifications to the published protocol, please also include them.

Reviewers' comments:

Reviewer #1 (Remarks to the Author):

The manuscript by Napso et al identifies proteins secreted by the placenta that could potentially be used as disease pathology markers. This work was initiated using two complementary murine approaches to narrow in on proteins secreted by endocrine-type cells in the junctional zone. Cultured trophoblasts, FACS sorting, and conditioned media were used to enrich these cells for LC/MS-based proteomics. Pathway analyses were used to determine signaling networks that were observed in these models and when compared to known proteins found in human placentas (healthy and complicated pregnancies). This list was narrowed to 4 proteins that were subsequently quantified in the circulation of subjects with gestational diabetes. The paper provides a unique and exciting approach to identify candidate proteins. The paper is also relatively straight forward to follow and well-written.

Major Concerns:

1. In Figure 1, the authors were careful to quantify cell death and observed a peak at 24 h. However, the proteomics data were conducted on media collected between 24-48 h. This raises concern that subsets of the proteins in the media may reflect ex vivo cell death (such as DAMPs) associated with culturing of primary cells rather than secreted proteins involved in cell-cell communication or maintenance of placental health. Were any of the secreted proteins from Figure 1G (conditioned media) known DAMPs?

There has been a mis-understanding. To clarify, the conditioned media was collected at 48h, not 24-48h (please see line 643). We specifically chose this time point, to allow the cells to stabilise to the culture conditions. In fact, at this time point the level of cell death as indicated by LDH and p53 / Bax expression approached values seen at 0h. In total, there are 24 potential DAMPs (1) in the list of 319 proteins that define the “placental secretome”. However, we and others have detected 19 of those potential DAMPS in the plasma of mice (see table below). Thus we believe that at least these putative DAMPs produced by the placenta could be involved in cell-cell communication and maintenance of placental health. We have included details about this about this in the revised manuscript. Please see new Table S3 and lines 202-204 and 366-370.

Table of putative DAMPs in the 319 proteins that define the “placental secretome”

DAMP proteins in 319 proteins define the “placental secretome”	Detected in mouse plasma in this study	Detected in non-pregnant mouse plasma by others	Detected in the sorted cells
PGS1_MOUSE	No	No	No
CALR_MOUSE	No	Yes(2)	Yes
PGS2_MOUSE	Yes	Yes(2,3)	No
FINC_MOUSE	Yes	Yes(2,3)	No
GPC1_MOUSE	No	Yes(2)	No
H2B1C_MOUSE	No	No	Yes
H2B1F_MOUSE	No	Yes(2)	Yes
H2B1K_MOUSE	No	No	Yes
HMGB1_MOUSE	Yes	Yes(2)	No
HS90A_MOUSE	No	Yes(2)	Yes
HS90B_MOUSE	No	Yes(2,3)	Yes
ENPL_MOUSE	No	Yes(2)	Yes
HSP74_MOUSE	Yes	Yes(2)	Yes
BIP_MOUSE	No	Yes(3)	No
HSP7C_MOUSE	No	Yes(2,3)	Yes
HS105_MOUSE	No	No	Yes
IL1R2_MOUSE	No	Yes(2)	No
PPIA_MOUSE	Yes	Yes(2)	Yes
S10AA_MOUSE	No	No	Yes
S10AB_MOUSE	Yes	Yes(2,3)	Yes
S10A6_MOUSE	Yes	Yes(2)	Yes
S10A9_MOUSE	Yes	Yes(2,3)	No
SCRB2_MOUSE	No	Yes(2)	No
TENA_MOUSE	No	Yes(2,3)	No

References:

1. Roh JS, Sohn DH. Damage-Associated Molecular Patterns in Inflammatory Diseases. *Immune Netw.* 2018;18(4):e27.
2. Yang YR, Kabir MH, Park JH, Park JI, Kang JS, Ju S, Shin YJ, Lee SM, Lee J, Kim S, Lee KP, Lee SY, Lee C, Kwon KS. Plasma proteomic profiling of young and old mice reveals cadherin-13 prevents age-related bone loss. *Aging (Albany NY).* 2020;12(9):8652-8668.
3. Michaud SA, Sinclair NJ, Petrosova H, Palmer AL, Pistawka AJ, Zhang S, Hardie DB, Mohammed Y, Eshghi A, Richard VR, Sickmann A, Borchers CH. Molecular phenotyping of laboratory mouse strains using 500 multiple reaction monitoring mass spectrometry plasma assays. *Commun Biol.* 2018;1:78.

2. After identifying the secreted proteins from Figures 1 and 2, did the authors evaluate the relationship of the overlapping proteins in the two models with serum from non-pregnant and pregnant mice? This would help to refine the analysis to placenta-secreted/enriched in the circulation.

As recommended by the reviewer, we have performed a screen of mouse plasma collected from pregnant (day 16 of gestation) and non-pregnant females and found that 45 of the proteins in our placental secretome map were detectable. Of these, ~50% (24 out of 45) showed a higher relative abundance in pregnancy between 1.5 and 50-fold increase compared to non-pregnant state (see figure 1a below). The other ~20 secreted proteins were unchanged or reduced in the maternal circulation during pregnancy, which reinforces the idea that the placenta may not be the only or the main source of these secreted proteins. A further seven secreted proteins exclusively expressed by the mouse placenta were also identified and highly enriched in the plasma of mice during pregnancy, as expected (see figure 1b below). Interestingly, several prolactins were detectable in the plasma of pregnant mice, which is remarkable given that 99.9% of proteins in the circulation are comprised of ~20 high abundant plasma proteins, severely limiting the ability to detect low abundant proteins. In addition, given that the mass spectrometry analyses are not capable of exhaustively detecting all proteins, we cannot exclude lack of pregnancy-related changes for any of the other 319 proteins in our placental secretome map. Hence, further verification is needed in the future. Please see text (lines 217-222, 370-377, 585-591, 709-721 and 738-741) and Figure 3F and Supplementary Figure 3 of the revised paper.

Figure 1a

Figure 1b

Figure 1. Heat map generated using Heatmapper showing the relative abundance of peptides in our ‘placenta secretome map’ (a) and secreted by the mouse placenta (b) (MS peak areas normalized to the bovine insulin internal standard), which are detected in non-pregnant (n=3) and pregnant mouse plasma (on day 16 of pregnancy, n=8).

- There are concerns with the presentation of the data for the 4 proteins circulating in human pregnancies at 12 and 28 weeks between healthy and GDM. When reviewing the raw data in Supplemental Figure 4, there were no statistical differences between healthy and GDM for any of the proteins quantified. There may be a trend for MIF but no p value is provided. The results section does not acknowledge that there were no differences when comparing absolute values. Then, the authors decided to use two variations on data reporting: 1) multiple of the median (MoM) and 2) ratios. While use of MoM in reproductive sciences was initiated to compare AFP values between laboratories, it does not appear to be warranted in this study as all of the samples were presumably run at the same time on the same ELISAs. Further, in Figure 4D, the authors express FLT and ANGPT2 relative to MIF levels and cite the use of ratios for sFlt, etc to justify this expression. In this case, it appears that the MIF values are what is actually driving the differences in ratios. This is not acknowledged. Likewise, this fact greatly impacts the FLT1 data as the enrichment of the two proteins in the circulation differs by a couple of log units and the ratio places greater weighting as a result on the MIF levels. While it is admirable to explore novel ways to express biomarker data, the fact that there are actually no differences in concentrations across the groups (and that this is not acknowledged) is concerning.

We understand the reviewer's points. However, many papers use MOM values for measuring placental/fetal derived proteins in the maternal circulation (for e.g., references (4-13) below). As requested, we have added the details of the p values to supplementary Figure 4 (as well as Figure 4, for consistency). We have also included some text into the revised paper to acknowledge that the absolute values for the proteins quantified were not statistically significant with our small sample size and that significance which we observe with MoM values may be because this calculation could place greater weighting on the MIF results. Please see revised Figure 4 and Supplementary Figure 4, as well as, lines 488-492.

Revised Figure 4

Revised Supplementary Figure 4

Minor Concern:

- As noted by Jackson labs, there is no such thing as a C57BL/6 mouse. More details regarding the substrain(s) used need to be included.

The mice we used were on a C57BL/6J genetic background. We included these details into the revised paper. Please see line 572.

Reviewer #2 (Remarks to the Author):

In the manuscript "Unbiased placental secretome characterization identifies candidates for pregnancy complications", Napso and colleagues sought to characterize the secretory output by the endocrine cells isolated from mouse placentas using different molecular methodologies and bioinformatic analyses, and then determined if these findings may potentially be relevant to human population by investigating a few of the identified secreted placental proteins in women with GDM. There are very few studies that examine the endocrine functions of the placenta and how this affects pregnancy outcomes or helps predict pregnancy outcomes. Furthermore, the bioinformatic pipelines used by the authors are novel, and the idea of generating a placental secretome map for outcome prediction or diseases diagnosis is also innovative; these methodologies may be applicable to other research areas outside placenta/pregnancy/reproduction. Hence, this work is significant and likely to be of high interest to a wide range of readers/researchers. The authors present their methods and results clearly, and are thoughtful in addressing some of the potential problems related to cell culture. Overall, the manuscript is well-written. I have only a few questions and minor suggestions to the authors.

1) My biggest concern with this manuscript is that the authors may have performed both mouse studies and the human validation study without considering the impact of fetal sex. Can the authors provide a rationale for why fetal sex is not considered? Will fetal sex have any influences on the placental secretome and will the resulting map generated from male placental endocrine cells vs female placental endocrine cells differ? Can't fetal sex be included as a variable in the bioinformatic analyses?

We consider sex differences in placental function and pregnancy outcomes to be of clinical relevance and agree that fetal sex may be an important modifier of the placental secretome. However, unfortunately, we did not collect information on fetal sex and whole litters of mouse placentas were used in both the cell culture and cell sorting experiments (we would assume our data represents 50:50 female:male). In mice, even if sex-specific differences do exist in the

placenta secretome, these hormones will mix-up in the maternal circulation and contribute together to changing maternal physiology. Moreover, the published mouse and human placenta databases (Table S1 and S7) used to verify our placental secretome map did not consider fetal sex. Hence, our placental secretome map is for both sexes collectively. Fortunately, in the preliminary human validation study, we have a similar proportion of each sex in the healthy cohort (n=10; female to male = 5/5) and in the GDM pregnancy (n=6; n=10; female to male = 2/4). Due to the small sample size and lack of access to more samples, we were unable to split the validation cohort to provide a definitive assessment of sex differences in the proteome in human pregnancy. However, we are currently working on other projects, which may be able to elucidate this area further. We have highlighted that addressing the effect of fetal sex on placental secretome should be a key area for future work in the revised discussion. Please see revised Table 1 and text in lines 551-557.

2) Although the bioinformatic pipeline and the idea of a placental secretome map are novel and exciting, many of the markers (such as FLT1, IGF2) or functional pathways (such as immune system, growth factor regulation) identified and reported in this manuscript have been previously suggested or shown to be associated with adverse pregnancy outcomes. It is unclear what new insights are gained and learned from generating a placental secretome map using such a sophisticated approach. Can the authors please comment about this in their discussion?

It is indeed correct that many of the markers or functional pathways identified using our approach have been previously suggested or shown to be associated with adverse pregnancy outcomes. This validates our methodological approach, as well as the findings of others that such markers / pathways may indeed be involved in the pathogenesis or serve as diagnostic indicators of those pregnancy complications, thereby reinforcing further work on them. However, it is important to note that several previously unrecognised or understudied markers and pathways were also identified using our approach (e.g. IDE and TTHY as unrecognised placental hormones potentially involved in placental-maternal communication, and unfolded protein response for GDM and neutrophil degranulation for PE), which provides further value and could be tested more specifically in future work. We have included sentences related to this in the revised paper. Please see lines 537-545.

3) Related to the second comment above - although it is true that the secretome map may enable the identification of new secreted proteins, their clinical values and diagnostic accuracy are still largely

unclear and remain to be thoroughly investigated. In this context, it is unclear if such information from the secretome map is of clinical relevance. Additionally, if data on multiple secreted proteins or a network of proteins are needed to improve the accuracy of diagnosis (as suggested by calculating the protein ratios), the feasibility of doing so in clinical settings - particularly low resource clinical settings - is unclear. Can the authors please comment about this as well in their discussion?

We acknowledge that most hospital laboratories do not have access to detailed proteomic analysis, but many identified proteins can be measured in other ways, for example, using immunoassays, which could be automated, facilitating widespread availability. Techniques such as multiplexing also make it feasible to run multiple related assays simultaneously. Furthermore, the antenatal screening programme (first or second trimester) is a good example of the ability of multiple measured biomarkers to identify high-risk pregnancies, with improved performance using a panel of biomarkers over a single test. While this may not be immediately accessible to hospital laboratories in lower-resource settings, many countries run an antenatal screening programme already, and integration of new test panels into that existing infrastructure is feasible. The aim of this study was not to provide new clinically validated biomarker panels ready for diagnostic/predictive use, but rather to provide a resource to support future development of biomarkers or biomarker panels, which may be very valuable in multiple disease areas. We therefore consider our results to have direct clinical relevance, but would consider further biomarker work up and validation to be beyond the scope of this study. We have included a comment about these aspects in the revised discussion. Please see lines 501-516.

4) The authors provided a rationale for the studied time point (gestational day 16 in mouse pregnancy), which is reasonable. It will also be a good idea to explain this mouse gestational time point in the context of human pregnancy, especially related to the time points examined in their human validation study (ie: week 12 and week 28).

It is hard to directly compare mouse to human pregnancy given the different tempo of fetoplacental development. However, crudely, day 16 of mouse gestation may be considered the start of the last trimester of human pregnancy. We realise that this late stage of mouse pregnancy does not relate at all to weeks 12 or 28 of human pregnancy, and this may explain why our preliminary proof of concept human validation study did not provide further promising results (although

sample size was still small). However, we specifically chose weeks 12 and 28 of for our human validation study as these time-points correspond to before and after/around the time a women would be screened for a pregnancy complication like GDM. We have provided some comment about this in the revised paper. Please see lines 497-501.

5) If the LCMS quantification method has been published, please include the reference (Lines #612 and 632). If there are any modifications to the published protocol, please also include them.

Apologies for this omission. Find corresponding reference now inserted into the revised paper. See line 681 and 702.

References

4. Liu HQ, Wang YH, Wang LL, Hao M. Predictive Value of Free beta-hCG Multiple of the Median for Women with Preeclampsia. *Gynecol Obstet Invest.* 2015.
5. Bestwick JP, Huttly WJ, Wald NJ. Serum marker truncation limits in first trimester antenatal screening for trisomy 18. *J Med Screen.* 2018;25(4):169-173.
6. Black C, Al-Amin A, Stolarek C, Kane SC, Rolnik DL, White A, da Silva Costa F, Brennecke S. Midpregnancy prediction of pre-eclampsia using serum biomarkers sFlt-1 and PlGF. *Pregnancy Hypertens.* 2019;16:112-119.
7. Bornstein E, Gulersen M, Krantz D, Cheung SW, Maliszewski K, Divon MY. Microarray analysis: First-trimester maternal serum free beta-hCG and the risk of significant copy number variants. *Prenat Diagn.* 2018;38(12):971-978.
8. Cuckle H. Prenatal Screening Using Maternal Markers. *J Clin Med.* 2014;3(2):504-520.
9. Donovan BM, Nidey NL, Jasper EA, Robinson JG, Bao W, Saftlas AF, Ryckman KK. First trimester prenatal screening biomarkers and gestational diabetes mellitus: A systematic review and meta-analysis. *PLoS One.* 2018;13(7):e0201319.
10. Mohamad Jafari R, Masihi S, Barati M, Maraghi E, Sheibani S, Sheikhvatan M. Value of Pregnancy-Associated Plasma Protein-A for Predicting Adverse Pregnancy Outcome. *Arch Iran Med.* 2019;22(10):584-587.
11. Lakhi N, Gavind A, Noretta M, Jones J. Maternal serum analytes as markers of adverse obstetric outcome. *The Obstetrician & Gynaecologist.* 2012;14:267-273.
12. Ren F, Hu YU, Zhou H, Zhu WY, Jia LI, Xu JJ, Xue J. Second trimester maternal serum triple screening marker levels in normal twin and singleton pregnancies. *Biomed Rep.* 2016;4(4):475-478.
13. Vranken G, Reynolds T, Van Nueten J. Medians for second-trimester maternal serum markers: geographical differences and variation caused by median multiples-of-median equations. *J Clin Pathol.* 2006;59(6):639-644.

REVIEWERS' COMMENTS:

Reviewer #1 (Remarks to the Author):

The authors have been quite responsive to the initial reviews including the addition of new data.

Reviewer #2 (Remarks to the Author):

The authors have addressed all of my concerns in this revised version of the manuscript, and I have no additional comments or concerns. I believe that this article is ready to move forward into publication.